# Gene interfered-ferroptosis therapy for cancers

Jinliang Gao[1], Tao Luo [1] & Jinke Wang [1✉]

Although some effective therapies have been available for cancer, it still poses a great threat to human health and life due to its drug resistance and low response in patients. Here, we develop a ferroptosis-based therapy by combining iron nanoparticles and cancer-specific gene interference. The expression of two iron metabolic genes (*FPN* and *LCN2*) was selectively knocked down in cancer cells by Cas13a or microRNA controlled by a NF-κB-specific promoter. Cells were simultaneously treated by iron nanoparticles. As a result, a significant ferroptosis was induced in a wide variety of cancer cells. However, the same treatment had little effect on normal cells. By transferring genes with adeno-associated virus and iron nanoparticles, the significant tumor growth inhibition and durable cure were obtained in mice with the therapy. In this work, we thus show a cancer therapy based on gene interference-enhanced ferroptosis.

[1] State Key Laboratory of Bioelectronics, Southeast University, Nanjing, China. ✉email: wangjinke@seu.edu.cn

Although some effective therapies have been available for cancer, it still poses a great threat to human health and life due to its drug resistance and low response in patients. Therefore, various cancer therapies have been being developed to induce cancer cell death through one form of cell death such as apoptosis, necrosis, necroptosis, and pyroptosis[1]. Recently, ferroptosis has attracted increasing attention as another form of cell death. Ferroptosis was identified as an iron-dependent nonapoptotic cell death in 2012[2]. Subsequently, it was found that ferroptosis is triggered by iron-catalyzed lipid peroxidation initiated by nonenzymatic (Fenton reactions) and enzymatic mechanisms [lipoxygenases (LOXs)][3]. In the former case, hydroxyl radicals are directly generated by the Fenton reaction between ferrous iron ($Fe^{2+}$) and hydrogen peroxide and initiate nonenzymatic lipid peroxidation[4,5].

Since being found, ferroptosis has attracted considerable attention due to its potential role as a target for novel therapeutic anticancer strategies[6–8]. Because ferroptosis is a form of regulated cell death, the immune system may function in part through ferroptosis to prevent tumorigenesis[9]. More interestingly, it was found that ferroptosis can propagate among cells in a wave-like manner, exhibiting a potent killing effect on neighboring cells[10–12]. Another attractive point of ferroptosis is its potential to overcome drug resistance such as chemoresistance[13–19], and improve cancer immunotherapy[20]. Therefore, the antitumor effects of ferroptosis has been being rapidly and widely explored in variant cancers[21]. However, most of these studies focused on finding various ferroptosis inducers of compounds for inhibiting or depleting system $x_c^-$, GPX4 and CoQ10[22]. These compounds may be challenged by the same resistance problem as the traditional cancer drugs[20]. For example, IREB2 was identified as one high-confidence gene in the erastin-induced ferroptosis, however, when REB2 was silenced, the expression of iron metabolism-related genes would change, making cells resistant to ferroptosis[2,23,24].

Currently, iron oxide nanoparticles (IONPs) have been successfully used as MRI contrast agent in cancer diagnosis and anemia treatment[25–27]. However, IONPs have not yet been used in the cancer treatment. Nevertheless, many studies have shown that IONPs can release ferrous ($Fe^{2+}$) or ferric ($Fe^{3+}$) in acidic lysosome in cells. The released $Fe^{2+}$ participates in the Fenton reaction to generate toxic hydroxyl radicals (•OH), one of reactive oxygen species (ROS), to induce ferroptosis of cancer cells[11,28,29]. Therefore, the ferroptosis-driven nanotherapeutics for cancer treatment become increasingly attractive[30]. Moreover, it was found that nanoparticle-induced ferroptosis in cancer cells eliminates all neighboring cells in a propagating wave[10]. However, cells are sensitive to iron concentration, and a little fluctuation in concentration can cause a great response[31]. Due to the importance of iron to cells, cells have evolved out a set of mechanisms to maintain intracellular iron homeostasis[32]. The intracellular iron is thus under delicate regulation to keep iron homeostasis[33–35]. Cells are therefore capable to effectively store and export the intracellular excess iron ions released from internalized IONPs. For example, in previous study, we found significant up-regulation of transcription of important genes responsible for exporting intracellular iron ion in cells treated by a DMSA-coated $Fe_3O_4$ nanoparticle[36]. Therefore, the ferroptosis induced by a single IONPs treatment is too little to be utilized to treat cancer in clinics.

Recently, an interesting study reported an IONPs-induced ferroptosis that was enhanced by a genetic change. In the study, it was found that a nanoparticle iron supplement, ferumoxytol, has an anti-leukemia effect in vitro and in vivo in leukemia cell with a low level of FPN[33]. This antitumor effect resulted from the inability of the type of leukemia cell to export the intracellular $Fe^{2+}$ generated from ferumoxytol, which makes the type of leukemia cell susceptible to the elevated ROS generated by $Fe^{2+}$ through Fenton reaction. However, this study relies on the rare naturally occurred low expression of ferroptosis-related genes such as FPN, still not resolving the problem how to artificially lower the expression of these genes in tumors of patients, so that this IONPs-induced ferroptosis therapy can be applied to more wide types of cancers.

Nuclear factor-kappa B (NF-κB) is a sequence-specific DNA-binding transcription factor that is over activated in almost all cancers[37,38]. Therefore, to inactivate the protein, we designed a gene expression vector that can express an artificial microRNA targeting NF-κB RELA/P65 under the control of a NF-κB-specific promoter[39]. This promoter consists of a NF-κB decoy sequence and a minimal promoter and thus its transcriptional activation activity is dependent on the NF-κB activity in cells[39]. We therefore name it as DMP, meaning Decoy-Minimal Promoter. Our previous results verified that DMP could be used to realize cancer cell-specific gene expression by depending on the NF-κB over-activity in cancer cells[40,41]. By using the promoter, we developed two kinds of cancer gene therapy. One is a cancer immunotherapy induced by an artificial neoantigen displayed on cancer cell surface[40]. The other is a cancer gene therapy induced by cutting telomere via CRISPR/Cas9 or oncogenic mRNAs via Cas13a[41,42]. In these studies, we successfully controlled neoantigen and Cas9 or Cas13a expression selectively in cancer cells by the DMP promoter.

In this work, based on these previous studies, we deduce that inhibition of the genes exporting intracellular iron ions should be lethal to the IONPs-treated cancer cells, because the inhibition would prevent cell from exporting IONPs-produced iron ions, which can thus enhance IONPs-induced ferroptosis. Base on this speculation, we develop a cancer therapy named as gene interfered-ferroptosis therapy (GIFT) by combining the DMP-controlled gene interference tools with the DMSA-coated $Fe_3O_4$ nanoparticles (FeNP). By specifically knocking down expression of two iron metabolic genes, FPN and LCN2, with the DMP-controlled CRISPR/Cas13a and microRNA (miRNA) in cancer cells, the FeNP treatment induced dramatic ferroptosis in a wide variety of cancer cells that represent various hematological and solid tumors. However, the therapy has little effect on normal cells. By using both viral (adenovirus associated virus, AAV) and non-viral (PEI-modified $Fe_3O_4$ nanoparticles, FeNC) vectors, the growth of different xenografted tumors in mice were also significantly inhibited by the therapy.

## Results and discussion

**Conceptualization of GIFT**. The principle of GIFT is schematically illustrated in Fig. 1a. GIFT consists of a gene interfering vector (GIV) and FeNP. GIV is composed of a promoter named DMP and downstream effector gene. DMP is a NF-κB-specific promoter that consists of a NF-κB decoy and a minimal promoter (Fig. 1b). Because NF-κB is a transcription factor that is widely over-activated in cancers, the expression of effector gene can be activated in cancer cells by NF-κB binding to DMP (Fig. 1a). In contrast, the effector gene cannot be expressed in normal cells due to lack of NF-κB (Fig. 1a). Therefore, DMP is a cancer cell-specific promoter. When the GIV of DMP-controlled CRISPR/Cas13a or miRNA is transfected into cancer cells, Cas13a or miRNA can be expressed. The expressed Cas13a protein can associate with guide RNA (gRNA) transcribed from a U6 promoter to form Cas13a-gRNA complex. The expressed miRNAs can associate with RNA-induced silencing complex (RISC). Both Cas13a-gRNA and miRNA-RISC complexes can target mRNA of interest to knockdown the expression of target gene in cancer cells.

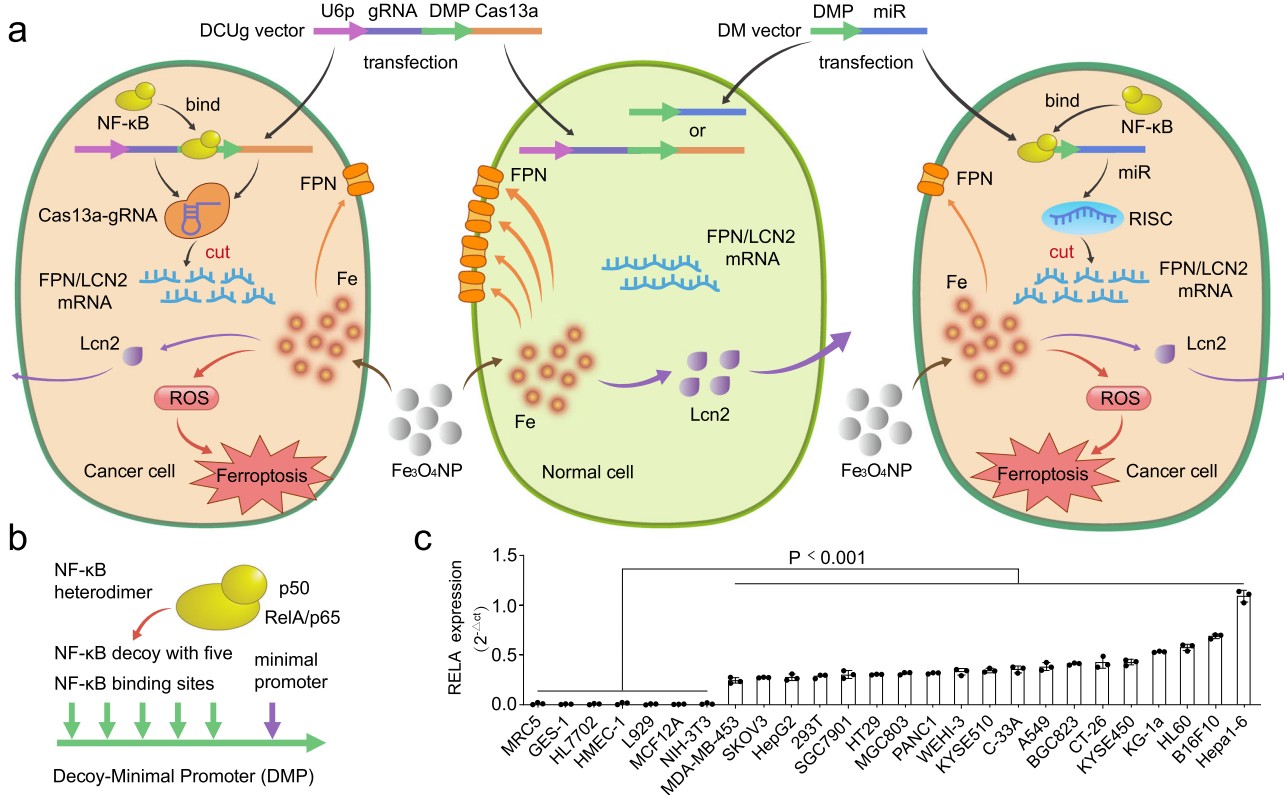

**Fig. 1 Schematic show of GIFT principle and NF-κB expression in variant cells. a** Schematic show of CRISPR/Cas13a- and miRNA-based GIFT. DCUg DMP-Cas13a-U6-gRNA, DM DMP-miRNA, U6p U6 promoter, gRNA guide RNA, DMP NF-κB decoy-minimal promoter, $Fe_3O_4$ NP $Fe_3O_4$ nanoparticle. **b** Schematic show of DMP and NF-κB dimer. **c** QPCR detection of NF-κB *RELA/P65* expression in various cells ($n = 3$ biologically independent samples). The statistical significance was obtained by comparing the data of cancer cells with normal cells. Data are presented as mean ± standard deviation (SD) and analyzed by one-way ANOVA with Tukey's test.

In this study, we selected two genes related to iron metabolism, *FPN* and *LCN2*, as target genes. The functions of both *FPN* and *LCN2* in cells are related to the efflux of iron ions[43–48]. In our previous study, we found the expression of the two genes were significant up-regulated when cells were treated with FeNP[36]. Therefore, we speculated that knocking down the expression of the two genes in cancer cells by the transfected GIVs of DMP-controlled CRISPR/Cas13a or miRNA would prevent cells from exporting the intracellular iron ions produced by the internalized FeNP. This can result in accumulation of iron ions and cause significant increase of intracellular ROS level, thus leading to significant ferroptosis in cancer cells. In normal cells, because Cas13a or miRNA cannot be produced, the expression of the two genes cannot be affected, which allows the normal cells to actively export the intracellular iron ions produced by FeNP and maintain the iron homeostasis, leading to little effects from FeNP treatment.

**Expression of NF-κB RELA in various cells**. NF-κB is widely activated in nearly all types of tumor cells. However, it is expected that its expression is not the same among cancer cells[49,50]. Because the intracellular NF-κB activity is critical to our strategy. Therefore, we first detected the expression of NF-κB *RELA/P65* in 3 leukemia cell lines (KG-1a, HL60 and WEHI-3), 16 solid tumor cell lines, and 7 normal cell lines (HL7702, MRC-5, HMEC-1, NIH-3T3, L929, GES-1, and MCF12A). The results demonstrated that NF-κB *RELA/P65* highly expressed at different levels in all cancer cells, but did not in all normal cells (Fig. 1c). Therefore, DMP should drive a cancer cell-specific gene expression.

**Effects of FeNP on cell viability**. To evaluate the cytotoxicity of FeNP that was stable at the long-time storage at 4 °C (Supplementary Fig. 1a), we dynamically measured the cell viability of three leukemia cells, a solid tumor cell (HepG2), and two normal human cells that were treated by various concentrations of FeNP for 5 days. The results showed FeNP had no significant toxicity to all cells below the dose of 50 μg/mL (Supplementary Fig. 1b). However, when the dosage was over 50 μg/mL, FeNP showed significant toxicity to all cells including normal cells (Supplementary Fig. 1b). Therefore, we used the FeNP at the concentration of 50 μg/mL for subsequent investigation, which is equivalent to the dose of intravenous injection of 3 mg/kg in rodents[33].

**Antitumor effect of GIFT in vitro**. To investigate the antitumor effects of GIFT, we first treated leukemia cells by GIFT. A human leukemia cell (KG-1a) was first transfected by various plasmid vectors and then treated by FeNP. The cell viability was detected by a acridine orange and ethidium bromide (AO&EB) dual staining at three time points. The results showed that only the combinations of all GIVs with FeNP caused the significant time-dependent cell death (Fig. 2a and Supplementary Fig. 2). All plasmids and FeNP alone and the combinations of negative control plasmids (pDCUg-NT and pDM-NT) with FeNP did not significantly affect cell viability (Supplementary Fig. 2). Especially, the co-expression of GIVs (pDCUg-FL and pDM-FL) showed the most significant cancer cell killing effect when combining with FeNP (Supplementary Fig. 2), suggesting a synergistic effect of co-interfering two genes. This same effect of

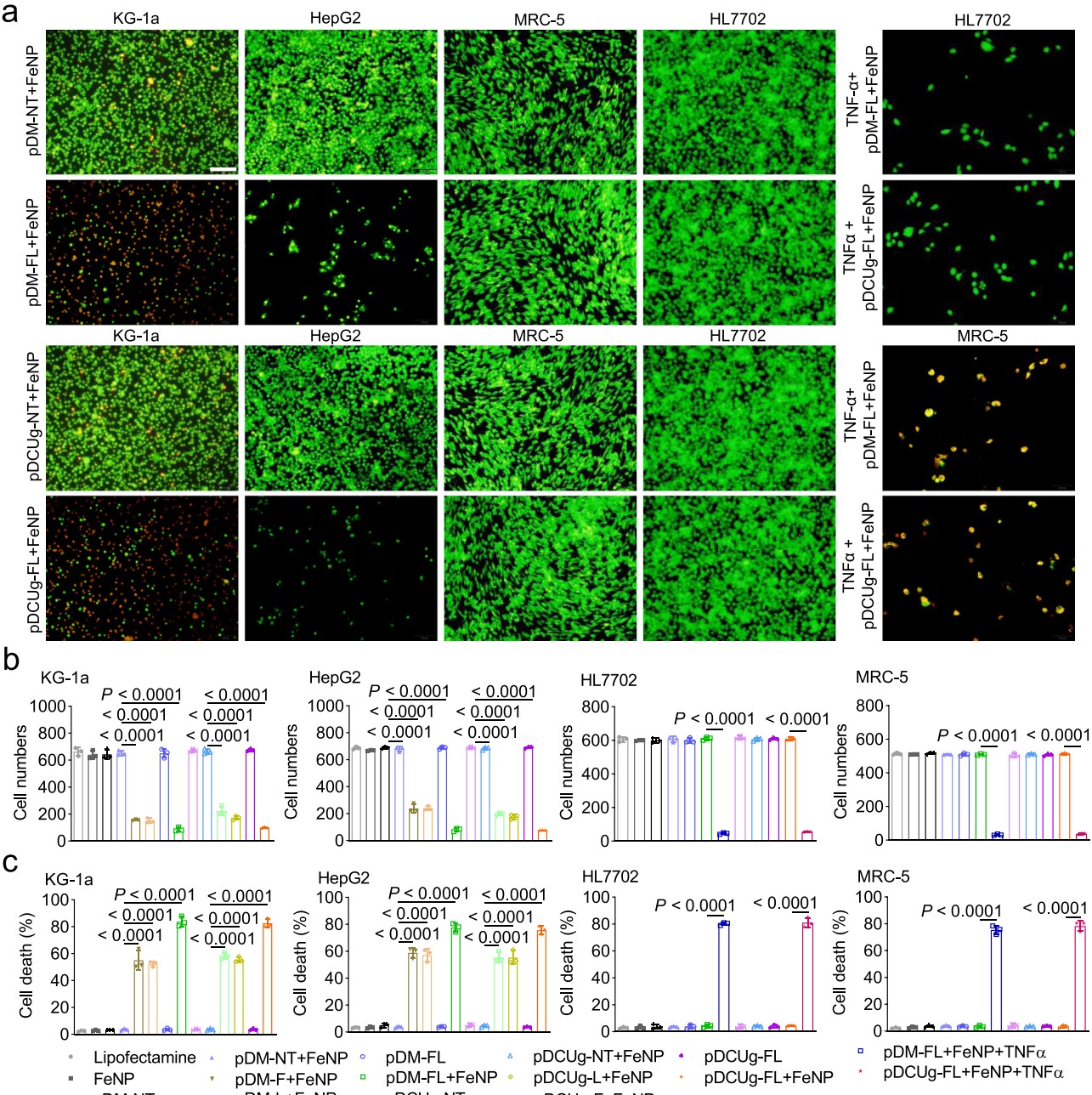

**Fig. 2 Treatment of tumor and normal cells with GIFT. a**, **b** Detection of cell viability. Cells (KG1a, HepG2, HL7702, and MRC5) were stained by AO&EB and imaged (**a**) and the images were quantified with ImageJ (**b**) ($n = 3$ independent micrographs). Scale bars in **a**, 100 μm. Only the representative images and quantified data of cells treated by pDCUg-FL, pDM-FL, pDCUg-NT, and pDM-NT together with FeNP for 72 h are shown here. The representative images and data of these cells treated by other reagents as more controls are shown in Supplementary Figs. 2, 3, 20, 21, 28, and 29. The images and quantified data of all other 22 cell lines are shown in Supplementary Figs. 4–19 and 22–29. **c** Detection of cell death ($n = 3$ biologically independent samples). The representative flow cytometer images are shown in Supplementary Figs. 30 and 31. Only the data of four cells (KG1a, HepG2, HL7702, and MRC5) are shown here. The data of other two cells (HL60 and WEHI-3) are shown in Supplementary Fig. 32. Data are presented as mean ± SD and analyzed by one-way ANOVA with Tukey's test. Plasmid used are as follows: pDCUg is pDMP-Cas13a-U6-gRNA; pDM is pDMP-miRNA; pDCUg-F and pDM-F target *FPN*; pDCUg-L and pDM-L target *LCN2*; pDCUg-FL and pDM-FL target both *FPN* and *LCN2*; pDCUg-NT and pDM-NT target no transcript. **b**, **c** use a same set of symbols.

GIFT was then observed on a human solid tumor cell HepG2 (Fig. 2a and Supplementary Fig. 3).

To investigate the broad spectrum of GIFT in killing cancer cells, we next similarly treated a variety of cancer cells representing different leukemia and solid tumors in human and mouse, including 2 leukemia cells (HL60 and WEHI-3) and 14 solid tumor cells (A549, HT-29, PANC-1, SKOV3, MDA-MB-453,

C-33A, BGC-823, SGC-7901, MGC-803, KYSE450, KYSE510, B16F10, Hepa1-6, and CT-26) with the co-expressed GIVs (pDCUg-FL and pDM-FL). The results revealed that the combinations of the two GIVs with FeNP produced the significant time-dependent killing effects in all tumor cells (Supplementary Figs. 4–19). Similarly, the vectors and FeNP alone and the combinations of negative control plasmids (pDCUg-NT and

pDM-NT) with FeNP did not significantly affect all cells at any treatment time (Supplementary Figs. 4–19).

To investigate the cancer cell specificity of GIFT, we next treated seven normal human and mouse cells (HL7702, MRC5, GES-1, HMEC-1, L929, MCF12A, and NIH-3T3). The results showed that all vectors alone and their combinations with FeNP did not significantly affect all these cells (Fig. 2a and Supplementary Figs. 20–26), which is consistent with low NF-κB expression in these cells (Fig. 1c). To further validate the key role of NF-κB activation in GIFT, we transfected these cells with pDCUg-FL and pDM-FL respectively and then induced them with a NF-κB activator, TNF-α. The cells were then treated by FeNP. We found that these cells were also significantly killed by GIFT with the TNF-α inducement (Fig. 2a and Supplementary Figs. 20–26). This confirms that cells can be killed by GIFT only when NF-κB is activated, which is also confirmed by the treatment of HEK-293T cell with GIFT. The HEK-293T cell is a human embryonic kidney cell that expresses large T antigen after being transfected with a virus. Although the cell is not considered as a cancer cell, its NF-κB expression is significantly activated (Fig. 1c). Therefore, the same GIFT effects as cancer cells were seen in this cell (Supplementary Fig. 27).

To further characterize the cell viability under the various treatments, we counted the numbers of alive cells in the AO&EB-stained images with ImageJ and identified statistical significance with GraphPad. The result showed that only the combinations of all GIVs with FeNP caused the significant time-dependent cancer cell death (Fig. 2b and Supplementary Figs. 28 and 29). After a treatment of 72 h, few cells remained alive, indicating the strong cancer cell killing effect of GIFT. The cell viability was further confirmed by detecting cell death, which also revealed the efficiency and specificity of GIFT leading cancer cells to die (Fig. 2c and Supplementary Figs. 30–32). The death detection also revealed the synergistic effect of co-interfering two genes (Fig. 2c and Supplementary Figs. 30–32).

To evaluate the interfering effect of DMP-Cas13a/U6-gRNA (DCUg) and DMP-miR (DM) systems, we next detected the expression of FPN and LCN2 genes in the treated cells. The results revealed that the expression of the two genes were significantly knocked down at both mRNA (Fig. 3a) and protein (Fig. 3b) levels by the targeting gRNAs/miRNAs in all detected cancer cells (KG-1a, HL-60, and HepG2). However, no significant changes were found in the HL7702 cell, further indicating the cancer cell specificity of the designed gene interfering systems. To further explore the cancer cell-specific expression of effector gene, we detected Cas13a mRNA in the treated cells. The results revealed that Cas13a only expressed in all cancer cells treated by pDCUg vectors (Fig. 3a).

**GIFT antitumor by ferroptosis**. It has been reported that iron-based nanomaterial can up-regulate ROS levels through the Fenton reaction[28]. To investigate whether ROS was produced in the FeNP-treated cells, we measured the ROS levels in four cells under the various treatments. The results revealed that the combinations of GIVs with FeNP resulted in the highest levels of ROS in all cancer cells (Fig. 3c and Supplementary Figs. 33 and 34a). However, the FeNP alone and the combinations of negative control vectors with FeNP only resulted in a little increase of ROS in all cancer cells (Fig. 3c and Supplementary Figs. 33 and 34a). In contrast, all the same treatments only resulted in a little increase of ROS in two normal cells (HL7702 and MRC5) (Fig. 3c and Supplementary Figs. 33 and 34a). Nevertheless, when two normal cells were induced by TNFα, the combinations of GIVs with FeNP immediately resulted in high ROS levels in two normal cells (Fig. 3c and Supplementary Fig. 33).

To further confirm the origin of ROS, we next measured the iron content in the treated cells. The results revealed that the intracellular iron content of four cancer cells significantly increased under the treatment of FeNP together with GIVs (Fig. 3d and Supplementary Fig. 34b). However, the FeNP alone and the combinations of negative control vectors with FeNP only resulted in a limited increase of iron content (Fig. 3d and Supplementary Fig. 34b). Similarly, when two normal cells were induced by TNFα, the combinations of GIVs with FeNP significantly resulted in high iron content in two normal cells (Fig. 3d). These data are consistent with the ROS levels in the four cancer cells under the same treatment. These data also show that the iron efflux was significantly inhibited by the knockdown of FPN and LCN2 in cancer cells (Fig. 3a, b), which leads to a great increase of intracellular ROS and iron contents (Fig. 3c, d).

To confirm the mechanism underlying GIFT is ferroptosis, we next treated the HepG2 cell with GIFT in the presence of various inhibitors including ferroptosis inhibitors (ferrostatin-1, Fer-1; liproxstatin-1, lipro-1), iron chelator (deferoxamine, DFO), cell-permeable analog of cysteine (N-acetylcysteine, NAC), apoptosis inhibitor (ZVAD-FMK, ZVAD), necrosis inhibitor (Necrostatin-1s, Nec1s), and autophagy inhibitor (Bafilomycin A1, BA1). The cell viability was detected with Cell-Titer-Glo 2.0. The results showed that the GIFT treatment significantly reduced the cell viability (Fig. 4a). The treatment of cells with Fer1, lipro-1, DFO, and NAC markedly reduced the GIFT-induced cell death, whereas inhibitors of apoptosis, necroptosis, or autophagy had little impact on cell death (Fig. 4a). Moreover, the cell viability was most significantly rescued by the cotreatment with Fer1, DFO, and NAC (FDN) (Fig. 4a). The similar results were also obtained in the cancer cells MDA-MB-453, CT-26, KG-1a, PANC1, and WEHI-3, and the TNFα-induced normal cells HMEC-1 and MRC-5 (Supplementary Fig. 35).

To further confirm the mechanism underlying GIFT is ferroptosis, we next detected the lipid ROS (a hallmark of ferroptosis) by using a lipid oxidation indicator, C11-BODIPY. It was found that only the GIFT treatment significantly induced lipid ROS in the HepG2 cell (Fig. 4b, c and Supplementary Fig. 36). Additionally, the increased lipid ROS could be reverted by Fer1, lipro-1, DFO, and NAC, but not by ZVAD, Nec1s, and BA1 (Fig. 4b, c and Supplementary Fig. 36). Moreover, the cotreatment with Fer1, DFO, and NAC (FDN) most strongly reverted the GIFT-induced lipid ROS (Fig. 4b, c and Supplementary Fig. 36). The similar results were also obtained in the cancer cells MDA-MB-453, KG-1a, WEHI-3, PANC1, and CT-26, and the TNFα-induced normal cells MRC-5 and HMEC-1 (Supplementary Figs. 37–44). The colony formation assay of four cell lines (HepG2, MDA-MB-453, CT-26, and PANC1) revealed that the GIFT treatment could nearly eradicate the colony formation capability of various tumor cells; however, the treatment of Fer1 and cotreatment of Fer1, DFO and NAC (FDN) could restore some colonies (Fig. 4d, e and Supplementary Figs. 45 and 46). These results indicated that the mechanism underlying GIFT is the designed gene-interfered ferroptosis.

**Virus-based GIFT antitumor in vivo**. To explore the in vivo antitumor effects of GIFT, we next cloned the GIVs into AAV to prepare recombinant viruses (rAAV). The rAAVs were tested by transfecting three cells (KG-1a, WEHI-3, and HL7702). The results showed that the combinations of gene interfering rAAVs (rAAV-DCUg-FL and rAAV-DM-FL) with FeNP resulted in significant death of two cancer cells (Supplementary Fig. 47). However, all rAAVs and FeNP alone and the combinations of negative control rAAVs (rAAV-DCUg-NT and rAAV-DM-NT) with FeNP did not significantly affect the

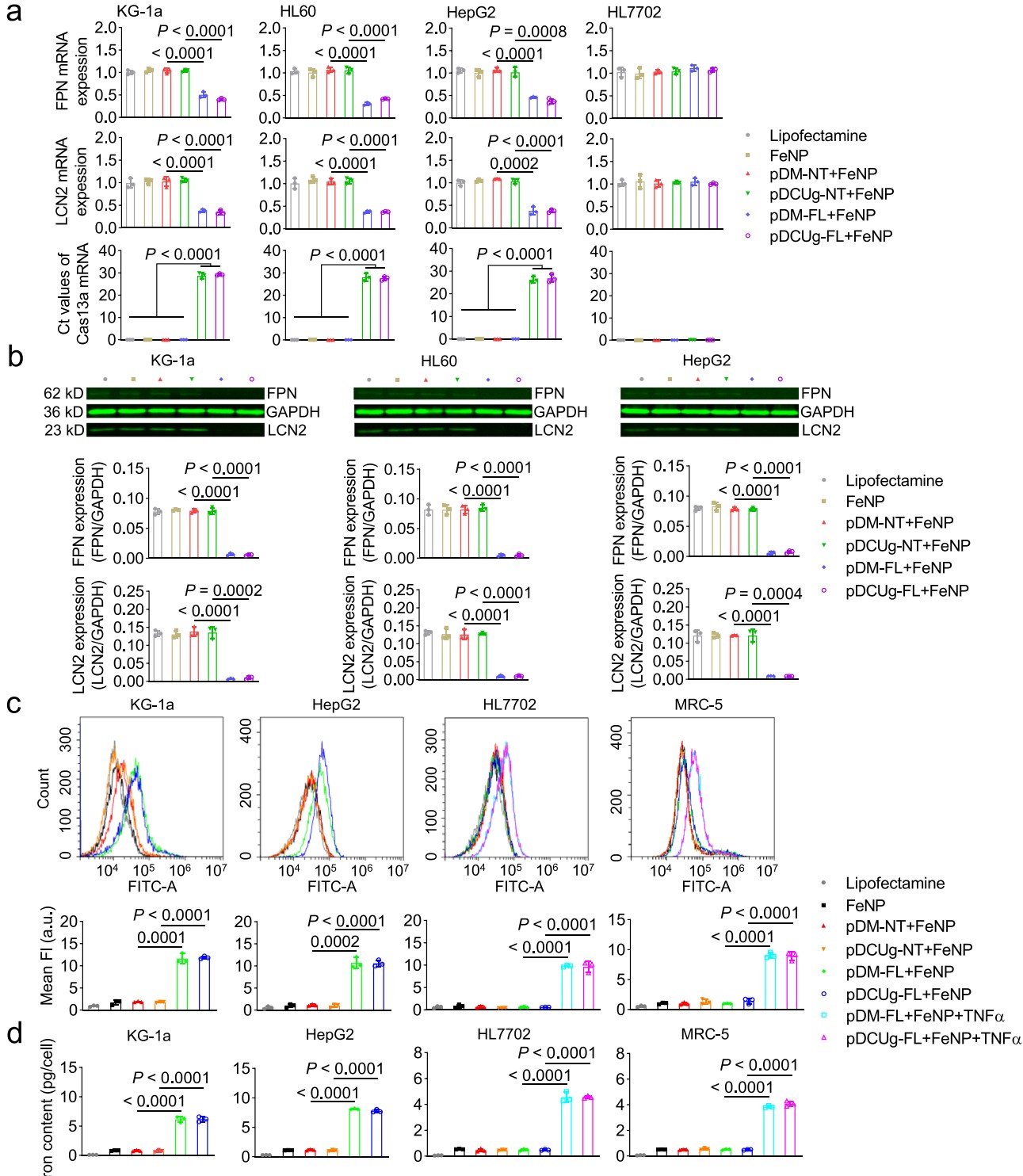

**Fig. 3 Gene expression, ROS production, and iron content in the GIFT-treated cells.** Cells were transfected by plasmid vectors and cultured for 24 h, and then incubated with or without 50 μg/mL of FeNP. Gene expression, ROS production, and iron content of cells were detected at 48 h post FeNP administration. **a** QPCR analysis of mRNA expression. **b** Western blot assay of protein expression. Samples were run on the same blot. The representative image and quantified optical density are shown. **c** Measurement of ROS levels. The fluorescence shift and quantified fluorescence intensity are shown. a.u., arbitrary units. **d** Measurement of iron content. **c**, **d** only show the data of four cell lines (KG1a, HepG2, HL7702, and MRC5). The representative images of flow cytometry detection of ROS are shown in Supplementary Fig. 33. The detection results of ROS and iron content of other two cells (HL60 and WEHI-1) are shown in Supplementary Fig. 34. **c**, **d** use a same set of symbols. Data are presented as mean ± SD ($n = 3$ biologically independent samples) and analyzed by one-way ANOVA with Tukey's test.

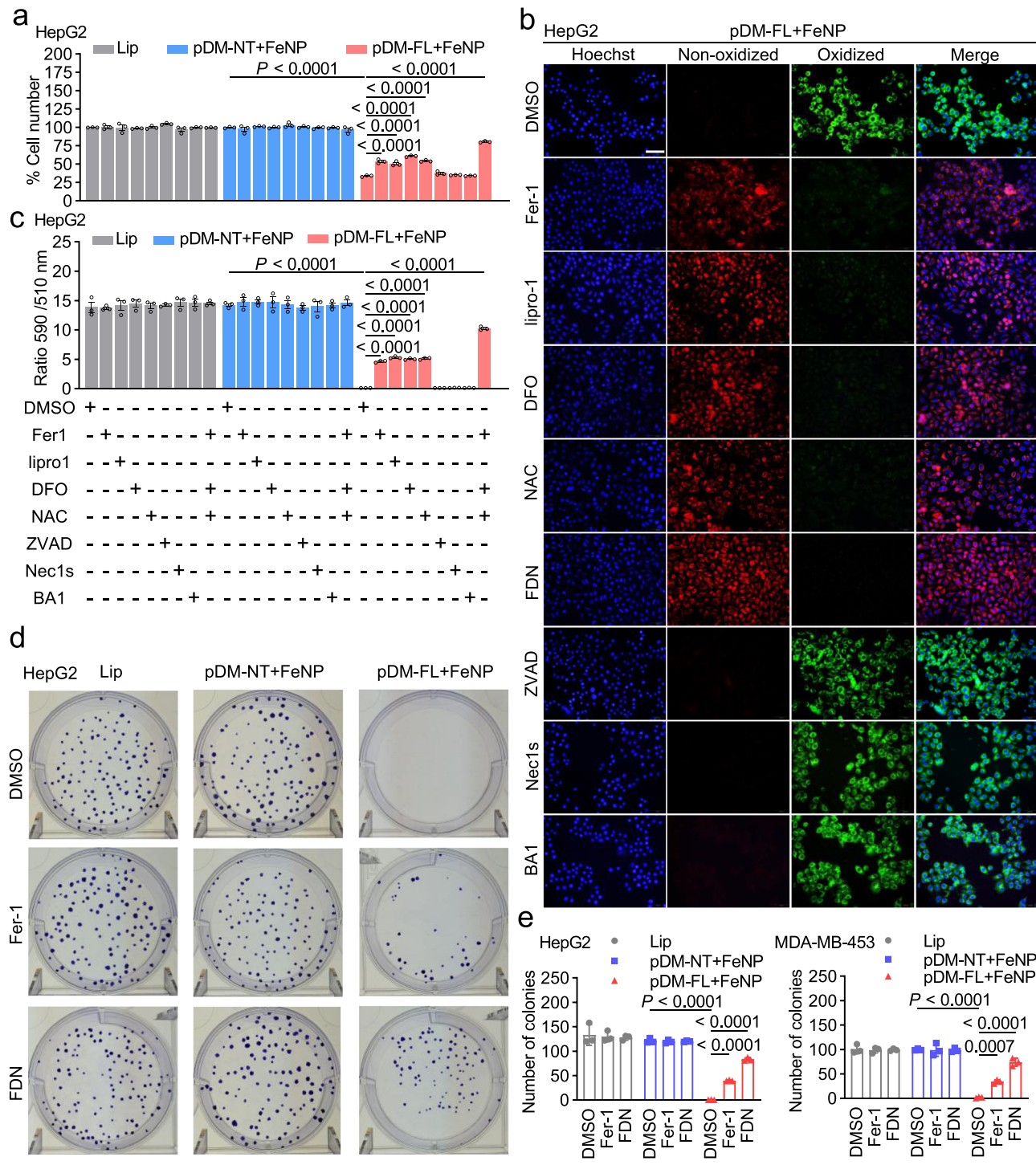

growth of all cancer cells (Supplementary Fig. 47). In human normal cell HL7702, all treatments caused no significant cell death (Supplementary Fig. 47).

To investigate the in vivo antitumor effect of GIFT, we next treated the cancer cell xenograft mouse with GIFT. The mouse leukemia cell WEHI-3 was subcutaneously transplanted into BALB/c mice to make tumor-bearing mice. Two batches of animal treatments were performed. The dosage of different viruses and FeNP were $1 \times 10^{10}$ vg/mouse and 3 mg/kg, respectively. In the first batch of animal treatment, six groups of tumor-bearing mice were treated with a single intravenous injection of separate rAAV and FeNP. The results showed that only the rAAV-DCUg-FL + FeNP treatment significantly

inhibited tumor growth (Supplementary Fig. 48a, b). In the second batch of animal treatment, five groups of tumor-bearing mice were treated with a single intravenous injection of mixed rAAV and FeNP. The results indicated that only the rAAV-DM-FL + FeNP treatment significantly inhibited tumor growth (Supplementary Fig. 48c, d). These results indicate that both separate and mixed intravenous injection can be used by GIFT. Therefore, to simplify the drug administration, we used the mixed intravenous injection in the following studies. To further characterize the in vivo antitumor effect of GIFT, we detected the abundance of DNA of GIV and mRNA of effector gene Cas13a and two target gene FPN and LCN2 in various tissues by qPCR. The results showed that the GIV DNA distributed in all

**Fig. 4 GIFT-induced cell death by ferroptosis.** Cells were transfected with various plasmids and cultured for 24 h. The cells were then co-incubated with FeNP and indicated inhibitors for 48 h. **a** Cell viability. The cell viability was detected by the Cell-Titer-Glo 2.0 reagent ($n = 3$ biologically independent samples). Only the results of the HepG2 cell are shown here. The results of other seven cell lines are shown in Supplementary Fig. 35. **b** Lipid ROS imaging. Lipid ROS production was detected by C11-BODIPY and imaged by fluorescence microscope. Scale bars, 50 μm. Only the representative images of the HepG2 cell that were treated by pDM-FL + FeNP and various inhibitors are shown here. The representative images of all eight cell lines under all various treatments are shown in Supplementary Fig. 36–43. Blue, nucleus; red, reduced dye; green, oxidized dye. **c** Lipid ROS quantification. Images of cells analyzed by ImageJ software and the ratio of intensity in 590 to 510 channels were calculated ($n = 3$ independent micrographs). Only the results of the HepG2 cell are shown here. Those of other seven cell lines are shown in Supplementary Fig. 44. **d**, **e** Colony formation. **d** Representative images of the HepG2 cell. All images of four cell lines are shown in Supplementary Figs. 45 (HepG2 and MDA-MB-453) and 46 (CT-26 and PANC1). **e** The quantified data of colony formation assay of two cells (HepG2 and MDA-MB-453) ($n = 3$ biologically independent samples). The quantified data of colony formation assay of the other two cells (CT-26 and PANC1) are shown in Supplementary Fig. 46. Each treatment was conducted in triplicates. Lip Lipofectamine, Fer1 Ferrostatin-1 (1 μM), lipro1 liproxstatin-1 (1 μM), DFO deferoxamine (100 μM), NAC N-acetylcysteine (1 mM), ZVAD ZVAD-FMK (50 μM), Nec1s Necrostatin-1s (10 μM), BA1 Bafilomycin A1 (1 nM), FDN co-incubation of Fer1, DFO and NAC. Data are presented as mean ± SD and analyzed by one-way ANOVA with Tukey's test.

detected tissues, especially in tumor (Supplementary Fig. 48e), and Cas13a was only expressed in tumors (Supplementary Fig. 48f). The expression of two target genes (*FPN* and *LCN2*) was only significantly knocked down in tumors (Supplementary Fig. 48g, h).

To improve the treatment effect, we subsequently treated tumor-bearing mice of WEHI-3 cell with multiple intravenous injections of rAAV-DM-NT/FL + FeNP ($1 \times 10^{10}$ vg/mouse plus 3 mg/kg FeNP). The tumor-bearing mice were intravenously administered three times with phosphate-buffered saline (PBS), rAAV-DM-NT + FeNP, and rAAV-DM-FL + FeNP (Supplementary Fig. 49a), respectively. The results indicated that the tumor growth was more significantly inhibited (Supplementary Fig. 49b). Moreover, the treatment with rAAV-DM-FL + FeNP significantly improved the survival of mice (Supplementary Fig. 49c). To further improve the treatment effect and check if tumor could be eradicated by GIFT, we next treated tumor-bearing mice of WEHI-3 cell with multiple intravenous injections of rAAV-DM-NT/FL + FeNP at a higher dosage ($2 \times 10^{10}$ vg/ mouse plus 3.6 mg/kg FeNP). The tumor-bearing mice were intravenously administered three times with PBS, rAAV-DM-NT + FeNP, and rAAV-DM-FL + FeNP (Fig. 5a), respectively. As a result, the rAAV-DM-FL + FeNP treatment more significantly inhibited tumor growth (Fig. 5b–e) and splenomegaly (Fig. 5f, g). The same treatment also more significantly prolonged the survival time of mice (Fig. 5h–k). However, no significant body weight change was observed during treatment in both mice used for measuring tumor inhibition (Fig. 5b) and survival (Fig. 5i), suggesting the good safety of GIFT reagents. Additionally, the addition of NAC to drinking water eliminated tumor inhibition (Fig. 5a–g) and survival improvement (Fig. 5h–k) of the rAAV-DM-FL + FeNP treatment. Furthermore, the in vivo antitumor effect of GIFT could be reverted by a more specific ferroptosis inhibitor, liproxstatin-1 (Supplementary Fig. 50). These data indicated that GIFT inhibits tumor by ferroptosis, consistent with the in vitro results. To further characterize the treatment, we also detected the iron content and the abundance of DNA of GIV and mRNA of *RELA*, *FPN,* and *LCN2* in various tissues. The results indicated that the rAAV-DM-FL + FeNP treatment significantly increased the iron content of tumors (Supplementary Fig. 51a), but not other tissues (Supplementary Fig. 51b). The GIV DNA distributed in all detected tissues, especially in tumor (Supplementary Fig. 51c). *RELA* was only highly expressed in tumors (Supplementary Fig. 51d) and two target genes FNP and LCN2 were only significantly knocked down in tumor (Supplementary Fig. 51e, f). Furthermore, the expression of two tumor growth markers, CD31 and Ki67, was also significantly downregulated by the treatment of rAAV-DM-FL + FeNP (Supplementary Fig. 51g, h),

which is consistent with the H&E staining of tumor sections (Supplementary Fig. 51i). Finally, the safety of the in vivo treatment was confirmed by the H&E staining of the main organs (Supplementary Fig. 52a) and detection of blood biochemical markers including white blood cells (WBC), red blood cells (RBC), platelet (PLT), and hemoglobin (HGB), hepatotoxicity markers including alanine aminotransferase (ALT), aspartate aminotransferase (AST), and alkaline phosphatase (ALP), and kidney injury markers including blood urea nitrogen (BUN), creatinine (Cr), and uric acid (UA) (Supplementary Fig. 52b).

To further confirm the observed antitumor effect of GIFT, we treated a new xenograft tumor model with GIFT. The tumor-bearing BALB/c mice were produced with the xenograft CT-26 cell, a widely used mouse colorectal cancer cell line. The tumor-bearing mice were intravenously administered three times with rAAV-DM-NT + FeNP and rAAV-DM-FL + FeNP at the same dose ($2 \times 10^{10}$ vg/mouse plus 3.6 mg/kg FeNP), respectively (Fig. 6a). The results revealed that the similar significant tumor growth inhibition (Fig. 6b–g) and survival improvement (Fig. 6h–k) were reproduced on the new tumor model. Detections of iron content of tumors (Supplementary Fig. 53a), abundance of rAAV DNA (Supplementary Fig. 53b), and mRNA of *RELA* (Supplementary Fig. 53c), *FPN* (Supplementary Fig. 53d), and *LCN2* (Supplementary Fig. 53e) obtained the similar results that were observed in the WEHI-3 model. The antitumor effects were also confirmed by the transcription of *CD31* (Supplementary Fig. 53f) and *Ki67* (Supplementary Fig. 53g) and the H&E staining of tumor sections (Supplementary Fig. 53h). The safety of the in vivo treatment was again confirmed by the H&E staining of the main organs (Supplementary Fig. 54a) and detection of blood biochemical markers, hepatotoxicity markers, and kidney injury markers (Supplementary Fig. 54b).

To determine whether the GIFT could induce antitumor effect to the metastatic tumors, we finally treated the lung metastatic melanoma formed by intravenously injected B16F10 cells in C57BL/6J female mice. The tumor-bearing C57BL/6J mice were intravenously administered three times with rAAV-DM-NT + FeNP and rAAV-DM-FL + FeNP at the same dose ($2 \times 10^{10}$ vg/ mouse plus 3.6 mg/kg FeNP), respectively (Fig. 7a). The results indicated that the rAAV-DM-FL + FeNP treatment significantly reduced tumor burden of lung (Fig. 7b–h). Similarly, rAAV-DM-FL + FeNP treatment significantly improved the survival of mice (Fig. 7i–k). Moreover, the rAAV-DM-FL + FeNP treatment significantly decreased splenomegaly (Supplementary Fig. 55a, b) and white blood cells (Supplementary Fig. 55c). Although the rAAV distributed in all tissues (Supplementary Fig. 55d), the GIFT reagents did not cause significant toxicity to the major organs (Supplementary Fig. 55e–h).

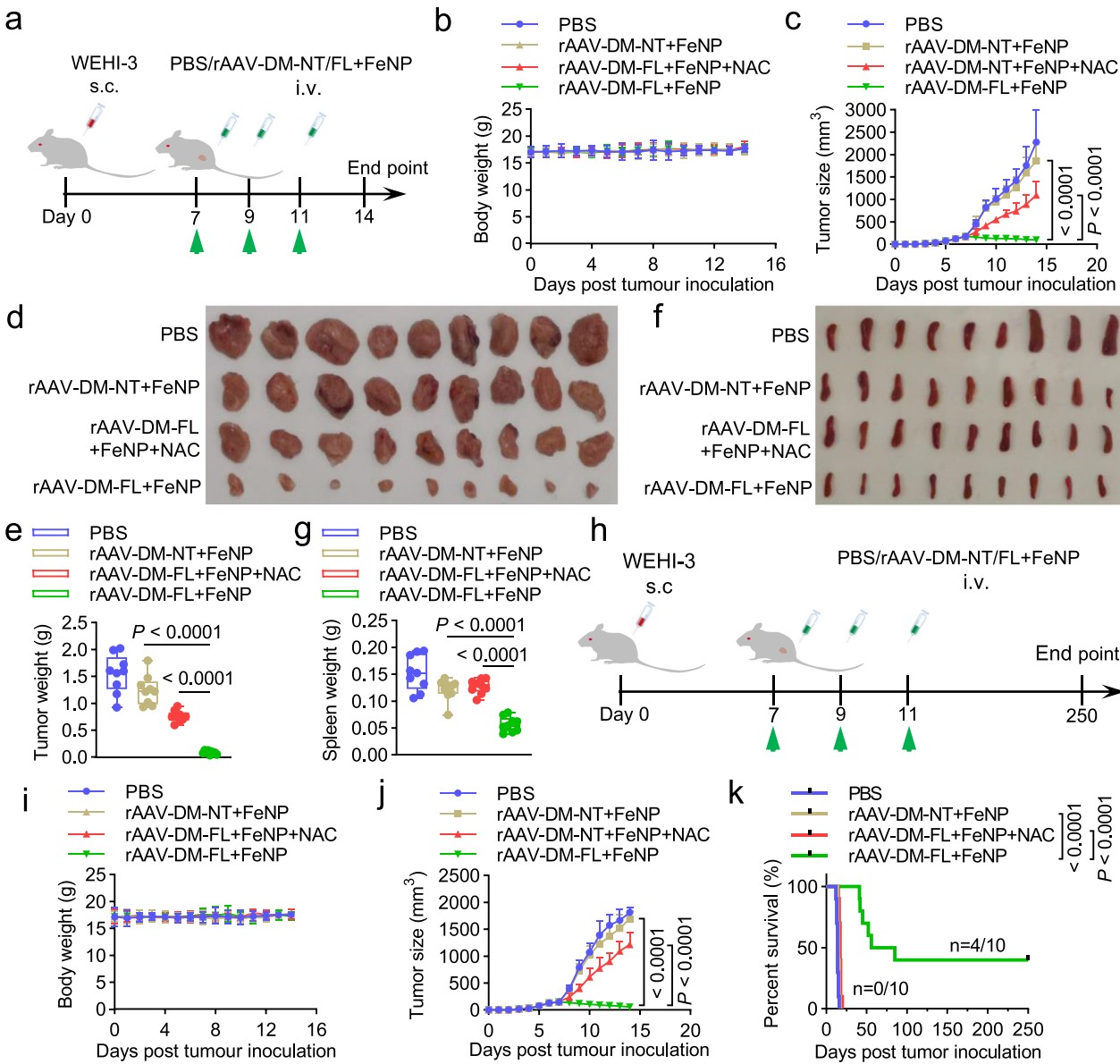

**Fig. 5 The in vivo antitumor effects of GIFT in the WEHI-3 xenograft mice.** The leukemia mice model was constructed by subcutaneously injecting the WEHI-3 cells. **a–g** Tumor growth detection. **a** Schematics of animal treatment. s.c. subcutaneous injection, i.v. intravenous injection. **b** Average body weight. **c** Tumor growth curve. **d** Tumor imaging. **e** Tumor weight. **f** Spleen imaging. **g** Spleen weight. The box-whisker plots in **e** and **g** show median (middle line), the 25th and 75th percentile (box) and min to max (whiskers). Other data are presented as mean ± SD ($n = 9$ mice). **h–k** Survival detection. **h** Schematics of animal treatment. **i** Average body weight. **j** Tumor growth curve. Data are presented as mean ± SD ($n = 10$ mice). **k** Kaplan–Meier survival curve ($n = 10$ mice). The statistical significance was analyzed by one-way ANOVA with Tukey's test in **c**, **e**, **g**, **j** and by the log-rank test in **k**.

To further confirm the safety of GIFT treatment of such multiple intravenous injection, we then treated two groups of healthy BALB/c mice with three times intravenous injection of PBS and rAAV-DM-FL + FeNP ($2 \times 10^{10}$ vg/mouse plus 3.6 mg/kg FeNP), respectively (Supplementary Fig. 56a). The results revealed that the rAAV-DM-FL + FeNP treatment did not cause significant loss of body weight (Supplementary Fig. 56b). Moreover, although the rAAV distributed in all tissues (Supplementary Fig. 56c), the GIFT reagents did not cause significant toxicity to the major organs (Supplementary Fig. 56d–g). These data together with those detected in tumor-bearing mice (Supplementary Figs. 52, 54, and 55) demonstrate the safety of the GIFT treatment.

**FeNC-based GIFT antitumor in vitro and in vivo.** Finally, to find whether iron nanoparticle itself can be used to transfer GIVs

as AAV, we selected a PEI-modified $Fe_3O_4$ nanoparticle (FeNC) as a DNA transfection agent. Two batches of FeNC (FeNC-1 and FeNC-2) were evaluated by two experiments. In the first experiment, plasmids were added to FeNC-1 to prepare FeNC-1@DNA. The KG-1a and HepG2 cells were first treated with FeNC-1@DNA (0.5 μg FeNC-1) just for transfecting GIVs and then treated with FeNP (50 μg FeNP). The results showed that only FeNC-1@pDM-FL/pDCUg-FL + FeNP resulted in significant cell death (Supplementary Figs. 57 and 58), whereas FeNC-1@pDM-FL/pDCUg-FL (0.5 μg FeNC-1) and FeNP alone and FeNC-1@pDM-NT/pDCUg-NT + FeNP did not significantly affect the cell growth (Supplementary Figs. 57 and 58). These data indicate that FeNC can be used as gene transfection agent in GIFT. In the second experiment, to simplify the GIFT reagents, we removed FeNP and just treated the KG-1a cell with FeNC@DNA. The

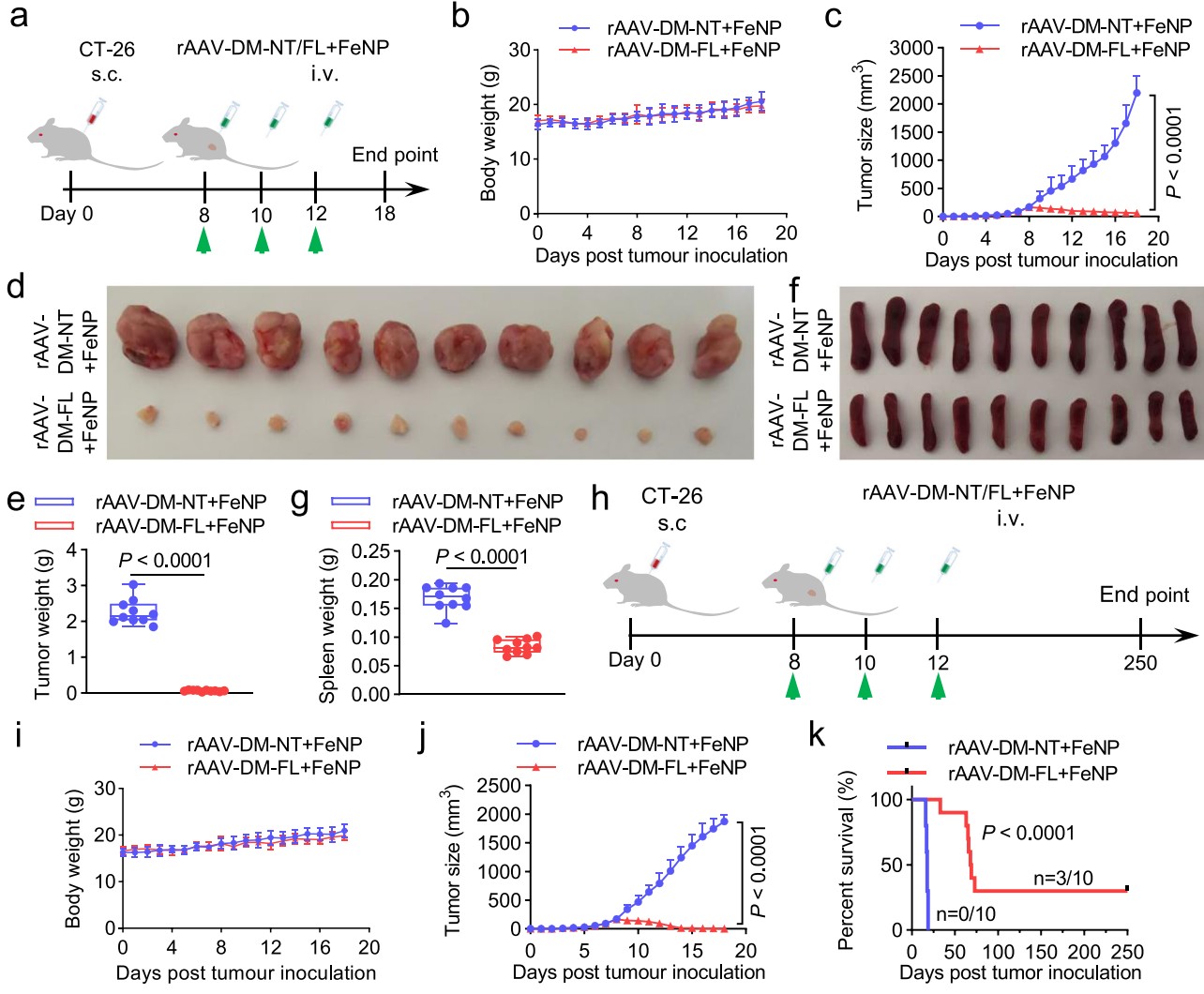

**Fig. 6 The in vivo antitumor effects of GIFT in the CT-26 xenograft mice.** The colon cancer mice model was constructed by subcutaneously injecting the CT-26 cells. **a–g** Tumor growth detection. **a** Schematics of animal treatment. s.c. subcutaneous injection, i.v. intravenous injection. **b** Average body weight. **c** Tumor growth curve. **d** Tumor imaging. **e** Tumor weight. **f** Spleen imaging. **g** Spleen weight. The box-whisker plots in e and g show median (middle line), the 25th and 75th percentile (box) and min to max (whiskers). Other data are presented as mean ± SD ($n = 10$ mice). **h–k** Survival detection. **h** Schematics of animal treatment. **i** Average body weight. **j** Tumor growth curve. Data are presented as mean ± SD ($n = 10$ mice). **k** Kaplan–Meier survival curve ($n = 10$ mice). The statistical significance was analyzed by a two-tailed Student's $t$ test in **c**, **e**, **g**, **j** and by the log-rank test in **k**.

results indicated that the treatment of FeNC@pDM-FL (50 μg FeNC) induced significant cell death (Supplementary Fig. 59), whereas FeNC (50 μg FeNC) and pDM-FL alone did not significantly affect the cell growth (Supplementary Fig. 59). To investigate the stability of FeNC@DNA, we also treated the KG-1a cell with the FeNC@pDM-FL kept for 24 h. The results showed that FeNC@DNA still had similar cancer cell killing effect (Supplementary Fig. 59). The followed lipid ROS and cell viability detections also indicated that FeNC@pDM-FL made cell death by ferroptosis (Supplementary Figs. 60 and 61).

To further investigate the antitumor effect of FeNC@DNA, we treated the tumor-bearing mice of WEHI-3 cell with a single intravenous injection of FeNC@pAAV-DCUg-FL/DM-FL and other reagents as controls. The results indicated that only the treatment of FeNC@pAAV-DCUg-FL/DM-FL induced significant tumor growth inhibition (Supplementary Fig. 62a, b). The detection of plasmid DNA revealed that the vector DNA distributed in all detected tissues (Supplementary Fig. 62c). The detection of Cas13a indicated that this effector gene was only expressed in tumors (Supplementary Fig. 62d). Moreover, the

transcription of two target genes *FPN* and *LCN2* were only significantly knocked down in tumors (Supplementary Fig. 62e, f). To further improve the antitumor effect of FeNC@DNA, we treated the tumor-bearing mice of WEHI-3 cell with two times intravenous injections of FeNC@pAAV-DM-FL and FeNC@-pAAV-DM-NT, respectively (Supplementary Fig. 62g). The results indicated that the tumor growth was more significantly inhibited (Supplementary Fig. 62h–j). The results also showed that the pAAV-DM-FL + FeNC treatment only significantly increased the iron content of tumors (Supplementary Fig. 62k).

**Pharmacokinetics and biodistribution.** Finally, we investigated the pharmacokinetics and biodistribution of rAAV-DM-FL, FeNP, pAAV-DM-FL, and FeNC. The results indicated that the concentration of rAAV-DM-FL, FeNP, pAAV-DM-FL, and FeNC in blood became approximately 9.91%, 5.41%, 2.7%, and 4.44% of the injected dose per gram of tissue (% ID/g) at 24 h post injection, respectively (Supplementary Fig. 63). The in vivo half-life time of them was 11.1 h, 5.0 h, 5.2 h, and 4.9 h, respectively (Supplementary Fig. 63). The detection of in vivo biodistribution

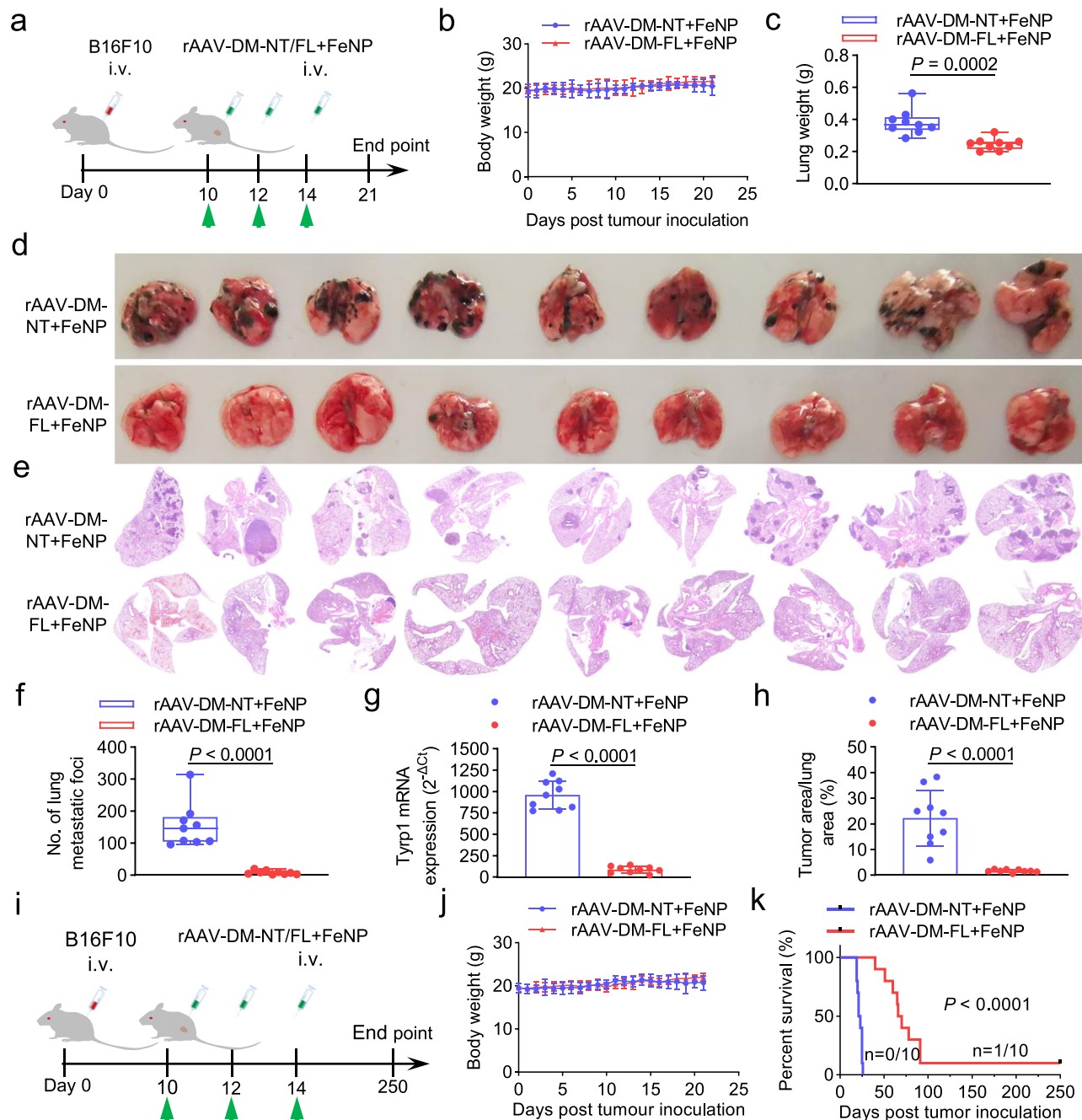

**Fig. 7 The in vivo antitumor effects of GIFT in the B16F10 xenograft mice.** The pulmonary metastatic melanoma model was constructed by intravenously injecting the B16F10 cells. **a**–**h** Tumor growth detection. **a** Schematic of animal treatment. i.v., intravenous injection. **b** Average body weight. **c** Lung weight. **d** Lung imaging. **e** H&E-stained lung section imaging. **f** Quantified B16F10 lung metastatic-like tumor foci. **g** QPCR detection of melanocyte-specific Tyrp1 mRNA expression in lung. **h** Percentage of tumor area to lung area in **e**. The box-whisker plots in **c** and **f** show median (middle line), the 25th and 75th percentile (box) and min to max (whiskers). Other data are presented as mean ± SD (*n* = 9 mice). **i**–**k** Survival detection. **i** Schematic of animal treatment. **j** Average body weight. Data are presented as mean ± SD (*n* = 10 mice). **k** Kaplan–Meier survival curve (*n* = 10 mice). The statistical significance was analyzed by a two-tailed Student's *t*-test in **c**, **f**, **g**, **h** and by the log-rank test in **k**.

of rAAV-DM-FL, FeNP, pAAV-DM-FL, and FeNC in tumors and major organs indicated that all these reagents were significantly sequestered by the reticuloendothelial system in the liver and spleen compared with other organs (Supplementary Fig. 63). The 29.5%, 24.3%, 20.4%, and 23.4% of rAAV, FeNP, pAAV, and FeNC (%ID/g) were accumulated in the tumors at 24 h post injection, respectively (Supplementary Fig. 63).

Sufficient concentration of systemically delivered iron oxide nanoparticles in tumors is important for GIFT. This study

demonstrates that two used iron nanoparticles can concentrate to tumors (Supplementary Figs. 51a, b and 63b, d) and GIFT thus significantly increased the iron content of tumors (Supplementary Figs. 51a, 53a and 62k). Because the used nanoparticles were not modified with any tumor-specific ligands, the nanoparticles should accumulate to tumors via passive enhanced permeability and retention effect (EPR) and active transport through trans-endothelial pathways[51]. The latter is recently reported accounting for the majority of nanoparticle accumulation in tumors[51].

Because FeNP is modified with no tumor-specific ligands, it can be generally used to treat three different kinds of tumors (Figs. 5–7). At present, few highly specific biomakers were identified on tumor cell surface, especially on solid tumors. Therefore, preparing nanoparticles with highly tumor specificity is still difficult. On the other hand, if a targeting ligand specific to a kind of tumor is modified on nanoparticles, this kind of nanoparticles may be unsuitable for treating other tumors because different tumors have different targets[52], which can complicate the preparation of nanoparticles for treating different tumors. Anyway, the nanoparticles modified with a tumor-specific ligand may improve the therapeutical effect of GIFT to the tumor by accumulating more nanoparticles. GIFT has its unique benefits by using iron oxide nanoparticles and AAV. Iron oxide nanoparticles are the only approved metallic nanoparticles for clinical use because of their demonstrated safety. The approved applications of iron oxide nanoparticles include cancer diagnosis, cancer hyperthermia therapy, and iron deficiency anemia[53]. Moreover, AAV is also an approved gene vector for clinical use and now more and more gene therapies employ it for its safty[54,55].

## Methods

**Vector construction.** The decoy minimal promoter (DMP), a chemically synthesized NF-κB-specific promoter which contains a NF-κB response sequence and a minimal promoter sequence, was cloned into pMD19-T simple (TAKARA) to obtain pMD19-T-DMP firstly, then the human codon optimized Cas13a coding sequence amplified from the pC013-Twinstrep-SUMO-huLwCas13a (Addgene) by PCR was cloned into pMD19-T-DMP to obtain pMD19-T-DMP-Cas13a, next the chemically synthesized U6 promoter sequence and the direct repeat sequence of guide RNA of Cas13a separated by BbsI restriction sites were cloned into pMD19-T-DMP-Cas13a, respectively, to generate pDMP-Cas13a/U6-gRNA backbone (referred to as pDCUg).

The guide RNAs (gRNAs) targeting no transcript (NT), human or murine ferroportin (FPN), and Lipocalin 2 (LCN2) were designed by CHOPCHOP (http://chopchop.cbu.uib.no/) (Supplementary Table 1). The complementary oligonucleotides containing a 28-bp gRNA target-specific region and two flanking BbsI sites were chemically synthesized (Supplementary Table 2) and annealed into double-stranded oligonucleotides, and then ligated into pDCUg. The ligation reaction (10 μL) consisted of 10 units of BbsI (NEB), 600 units of T4 DNA ligase (NEB), 1× T4 DNA ligase buffer, 1 nM double-stranded oligonucleotides, and 50 ng plasmid pDCUg. The ligation reaction was run on a PCR cycler as follows: 10 cycles of 37 °C 5 min and 16 °C 10 min, 37 °C 30 min, and 80 °C 5 min. The generated plasmids were named as pDCUg-NT, pDCUg-FPN (pDCUg-F), and pDCUg-LCN2 (pDCUg-L), respectively. Due to the difference between human and mouse gene, pDCUg expression vectors respectively targeted to the human FPN and LCN2 genes and mouse FPN and LCN2 genes were constructed. A plasmid co-expressing gRNAs targeting FPN and LCN2 simultaneously, named as pDCUg-FL, was also constructed.

The universal miRNA expression vector pDMP-miR was constructed based on pCMV-miR which was previously kept by our laboratory by replacing the CMV promoter with DMP promoter. The miRNAs targeting human or murine FPN and LCN2 were designed by BLOCK-iT™ RNAi Designer (https://rnaidesigner. thermofisher.com/rnaiexpress/) (Supplementary Table 1). Oligonucleotide pairs synthesized by Sangon Biotech (Shanghai, China) (Supplementary Table 3) were denatured and then annealed to obtain double-stranded DNA (dsDNA), which were then linked with the linear pDMP-miR vector cleaved with BsmBI. The generated miRNA expression vectors targeting the FPN and LCN2 genes were named as pDMP-miR-FPN (pDM-F) and pDMP-miR-LCN2 (pDM-L), respectively. Vectors were detected with PCR amplification and verified by DNA sequencing. Due to the difference between human and mouse gene, pDM expression vectors respectively targeted to the human FPN and LCN2 genes and mouse FPN and LCN2 genes were constructed. A plasmid co-expressing miRNAs targeting FPN and LCN2, pDMP-miR-FPN-DMP-miR-LCN2 (pDM-FL), was also constructed. The miR-NT fragment was synthesized according to the sequence of plasmid pcDNA™ 6.2-GW/EmGFP-miR-Neg and ligated into pDMP-miR, named pDMP-miR-Negative (referred to as pDM-NT) as a negative control vector.

The DCUg-NT/FL and DM-NT/DM-FL sequences were amplified by PCR from pDCUg-NT/FL and pDM-NT/DM-FL, respectively. By using the MluI (upstream) and XbaI (downstream) restriction sites, the PCR fragments were cloned into pAAV-MCS (VPK-410, Stratagene) to construct the pAAV-DCUg-NT/FL and pAAV-DM-NT/DM-FL vectors, respectively.

**Nanoparticles, cells, and culture.** The DMSA-coated $Fe_3O_4$ magnetic nanoparticle (FeNP) were provided by the Biological and Biomedical Nanotechnology

Group of the State Key Lab of Bioelectronics, Southeast University, Nanjing, China[56]. This FeNP was characterized by our previous study[36,57–60]. A magnetic transfection agent (MagTransf™) was purchased from the Nanjing Nanoeast Biotech co., ltd (Nanjing, China), which was referred to as FeNC in this study. Cells used in this research included KG-1a (human acute myeloid leukemia cells), HL60 (human promyeloid acute leukemia cells) and WEHI-3 (mouse acute mononuclear leukemia cells), HEK-293T (human fetal kidney cells), HepG2 (human liver cancer cells), A549 (human lung cancer cells), HT-29 (human colon cancer cells), C-33A (human cervical cancer cells), SKOV3 (human ovarian cancer cells), PANC-1 (human pancreatic cancer cells), MDA-MB-453 (human breast cancer cells), Hepa1-6 (mouse hepatoma cells), B16F10 (mouse melanoma cells), BGC-823/MGC-803/SGC-7901 (Human gastric adenocarcinoma cells), KYSE450/KYSE510 (human esophageal carcinoma cells), CT-26 (mouse colon cancer cells), HL7702 (human normal hepatocytes), HMEC-1 (human microvascular endothelial cells), L929 (mouse fibroblast), NIH-3T3 (mouse embryonic fibroblast), MRC5 (human embryonic fibroblasts), GES-1 (human gastric mucosal epithelial cells), and MCF-12A (human breast epithelial cells). KG-1a, SKOV3, HMEC-1, and MCF-12A cell lines were acquired from American Type Culture Collection (ATCC). HL60, WEHI-3, HEK-293T, HepG2, A549, HT-29, C-33A, PANC-1, MDA-MB-453, Hepa1-6, B16F10, BGC-823/MGC-803/SGC-7901, KYSE450/KYSE510, CT-26, HL7702, L929, NIH-3T3, MRC-5, and GES-1 cell lines were obtained from the cell resource center of Shanghai Institutes for Biological Sciences, Chinese Academy of Sciences. Three leukemia cell lines, KG-1a, HL60, and WEHI-3, were cultured in Iscove's Modified Dulbecco's Medium (IMEM) (Gibco). HEK-293T, HepG2, Hepa1-6, C-33A, PANC-1, MDA-MB-453, B16F10, MRC-5, L929, and NIH-3T3 cells were cultured in Dulbecco's Modified Eagle Medium (DMEM) (Gibco). A549, HT-29, SKOV-3, BGC-823/MGC-803/SGC-7901, KYSE450/KYSE510, CT-26, MCF-12A, and HL7702 cells were cultured in Roswell Park Memorial Institute (RPMI) 1640 medium (Gibco). HMEC-1 cells were cultured in MCDB-131 medium (Thermo Fisher) supplemented with 10 ng/ml epidermal growth factor (EGF) (Thermo Fisher) and 10 mM L-glutamine (Thermo Fisher). All media were supplemented with 10% fetal bovine serum (HyClone), 100 units/mL penicillin (Thermo Fisher), and 100 μg/mL streptomycin (Thermo Fisher). Cells were incubated at 37 °C in a humidified incubator containing 5% $CO_2$.

**FeNP cytotoxicity measurement.** To determine the optimal dosage of nanoparticles. The in vitro cytotoxicity of FeNP was performed by using the CCK-8 assay. KG-1a, HL60, WEHI-3, HepG2, HL7702, and MRC-5 cells were seeded in 96-well plates at a density of 5000 cells/well, respectively. Cells were cultivated overnight and treated with various concentrations (0, 30, 50, 100, 150, 200, and 250 μg/mL) of FeNP for various times. Each treatment was performed with six groups of cells and each group had four replicates. In all, 10 μL of Cell Counting Kit-8 (CCK-8) solution (BS350B, Biosharp) was added to each well at variant time points (0 d, 1 d, 2 d, 3 d, 4 d, and 5 d) post treatment. After incubating for another 1 h at 37 °C, the optical density at 450 nm was measured using a microplate reader (BioTek) and Gen5 software.

**Cell treatment by FeNP-based GIFT.** Cells were first transfected with plasmids using Lipofectamine 2000 (Thermo Fisher Scientific) according to the manufacturer's instructions. Briefly, cells ($1 \times 10^5$) were seeded into 24-well plates overnight before transfection. Cells were then transfected with 500 ng of various plasmids including pDCUg-NT, pDCUg-F, pDCUg-L, pDCUg-FL, pDM-NT, pDM-F, pDM-L, and pDM-FL. The mouse and human cells were transfected with vectors targeting to mouse and human genes, respectively. The transfected cells were cultured for 24 h and then incubated with or without 50 μg/mL of FeNP, and cells were cultured for another 72 h. For HL7702 and MRC5, cells were first induced with or without 10 ng/mL TNF-α (Sigma) for 1 h before FeNP treatment. At 24 h, 48 h, and 72 h post FeNP administration, all cells were stained with AO&EB following the manufacturer's instructions. Cells were imaged under a fluorescence microscope (IX51, Olympus) to observe numbers of live and dead cells, and living cell numbers were counted from the cell images by the Image-Pro Plus software. To quantify cell death, cells were collected at 72 h post FeNP administration and detected with the Annexin V-FITC/PI double staining Apoptosis Detection Kit (BD, USA) according to the manufacturer's instructions. The fluorescence intensity of cells was quantified with CytoFLEX LX Flow Cytometer (Beckman) and CytExpert software.

**Cell Titer Glo assay.** For viability assays for testing various inhibitors, cells were plated in a 96-well plate (100 μL per well) at a density of 5000 cells per well, allowed to seed overnight, and then transfected with 500 ng of various plasmids including pDM-NT and pDM-FL. The transfected cells were cultured for 24 h and then co-incubated with FeNP (50 μg/mL) and the indicated inhibitors including Ferrostatin-1 (Sigma, SML0583) (1 μM), liproxstatin-1(ApexBio, B4987) (1 μM), deferoxamine (DFO) (ApexBio, B6068) (100 μM), N-acetylcysteine (NAC) (Sigma, A9165) (1 mM), ZVAD-FMK (ApexBio, A1902) (50 μM), Necrostatin-1s (BioVision, 2263-1) (10 μM), Bafilomycin A1 (Sigma, B1793) (1 nM), and cells were cultured for another 48 h. The viability assays utilized the Cell-Titer-Glo 2.0 reagent (Promega, G9243) according to the manufacturer's instructions. Briefly, all wells from 96-well plates were

aspirated followed by the addition of 100 μL of 50% Cell-Titer-Glo 2.0 reagent 50% cell culture medium to each experimental well. Plates were incubated at room temperature with gently shaking for 10 min to promote adequate mixing. Luminescence was subsequently measured using a microplate reader (BioTek) and Gen5 software.

**ROS measurement.** Cells were treated with FeNP as previously described. Briefly, Cells were seeded at a density of $1 \times 10^5$ cells/ml medium per well in 24-well plates for overnight growth. Cells were then transfected with 500 ng of various plasmids including pDCUg-NT, pDCUg-FL, pDM-NT, and pDM-FL. The transfected cells were cultured for 24 h and then incubated with or without 50 μg/mL of FeNP, and cells were cultured for another 48 h. The treated cells were stained with 2′,7′-dichlorodihydrofluorescein diacetate (DCFH-DA) using the Reactive Oxygen Species Assay Kit (Beyotime) and BODIPY® 581/591 C11 using the Image-iT™ Lipid Peroxidation Kit (Thermo Fisher) according to the manufacturer's instructions. ROS changes indicated by fluorescence shift was analyzed on a CytoFLEX LX Flow Cytometer (Beckman) and CytExpert software or imaged by fluorescence microscope (IX51, Olympus) using traditional 590 nm and 510 nm emission filters with a 40X objective. The lipid peroxidation in cells were determined by quantitating the fluorescence intensities analyzed with ImageJ 1.51j8 software and calculating the ratio of intensity in 590 to 510 channels.

**Iron content measurement.** The average intracellular iron content and iron content of tissues were measured by ICP-MS (Agilent Technologies 7700, USA). The measurement procedure can be summarized as the following: (i) cells were treated with FeNP as previously described. Intracellular iron was determined at 48 h post FeNP administration. Cells were washed with PBS, collected by trypsinization, counted, and precipitated by centrifugation. Tissues were weighed and transferred to the 5 mL centrifuge tubes. (ii) The cell precipitation or tissues were then added a certain amount of 65% nitric acid and heated for complete digestion. (iii) Iron standard solutions (GSB 04-1726-2004, Beijing) with different concentrations (0, 0.1, 0.2, 0.5, 1, 2, and 5 μg/mL) were prepared to establish the standard curve for ICP-MS measurement. The intracellular iron content was reported as average iron content per cell. The iron content of tissue was reported as iron content per mg tissue. Each experiment was repeated in triplicates.

**Western blot assay.** Cells were seeded into six-well plates at a density of $2 \times 10^5$ cells per well and grown overnight. Cells in wells were transfected with 1000 ng of plasmids including pDCUg-NT, pDCUg-FL, pDM-NT and pDM-FL, respectively. At 48 h post transfection, the whole-cell extracts were prepared using a Total protein Extraction kit (BC3711, Solarbio, China) according to the manufacturer's instructions. The protein lysates (20 μg/sample) were resolved by SDS-PAGE and the target proteins were detected with Western blot (WB) using the antibodies as follows: GAPDH Rabbit monoclonal antibody (ab181602, Abcam, UK) (1:10,000 dilution), SLC40A1 Rabbit polyclonal antibody (ab58695, Abcam, UK) (1:10,000 dilution), and Lipocalin-2 Rabbit polyclonal antibody (ab63929, Abcam, UK) (1:10,000 dilution). The second antibodies were IRDye® 800CW Goat anti-Rabbit IgG (C80118-05, Licor) (1:10,000 dilution). The blots were detected and fluorescence intensity was quantified with the Odyssey Infrared Fluorescence Imaging System (Licor) and Odyssey software.

**Clone formation assay.** Cells were transfected with plasmids including pDM-NT and pDM-FL. The transfected cells were cultured for 24 h and then incubated with 50 μg/mL of FeNP and various inhibitors. After 48 h, the treated cells were washed, trypsinized, counted, replated in six-well plates at a density of 200 cells/well. Plates were monitored every day using a light microscope. When colonies of >50 cells are clearly visible, colonies resulting from the survived cells were fixed with 4% paraformaldehyde (Sangon Biotech, China), stained with 0.1% crystal violet (Sangon Biotech, China) and counted. Each assay was conducted in triplicates.

**Virus preparation.** HEK293T cells were seeded into 75 cm² flasks at a density of $5 \times 10^6$ cells per flask and cultivated for overnight. Cells were then co-transfected with two helper plasmids (pHelper and pAAV-RC; Stratagene) and one of the pAAV plasmids (pAAV-DCUg-NT, pAAV-DCUg-FL, pAAV-DM-NT, and pAAV-DM-FL) using Lipofectamine 2000 according to the manufacturer's instructions. Cells were cultured for another 72 h. The cells and media were collected and kept at −80 °C overnight. The cells and media were then incubated in 37 °C water bath for 2 h. This freeze-thaw process was totally repeated three times. The 1/10 volume of pure chloroform was added to the cell lysate and the mixture was vigorously shaken at 37 °C for 1 h. The mixture was added NaCl to a final concentration of 1 M and shaken until NaCl dissolved. The mixture was centrifuged at 15,000 revolutions per minute (rpm) at 4 °C for 15 min and the supernatant was collected. The supernatant was added PEG8000 at a final concentration of 10% (w/v) and shaken until PEG8000 dissolved. The mixture was centrifuged at 15,000 rpm at 4 °C for 15 min. The supernatant was discarded and the pellet was dissolved into PBS. DNase and RNase were added to a final concentration of 1 μg/mL to the dissolved pellet. The mixture was incubated at room temperature for 30 min. The mixture was extracted once with chloroform (1:1 volume) and the aqueous layer that contained the purified virus was transferred to

a new tube. Titers of AAVs were determined by qPCR using the primers AAV-F/R (Supplementary Table 4). Quantified viruses were aliquoted and kept at −80 °C for later use. The obtained viruses were named as rAAV-DCUg-NT, rAAV-DCUg-FL, rAAV-DM-NT, and rAAV-DM-FL.

**Virus evaluation.** KG-1a, WEHI-3 and HL7702 cells were seeded into 24-well plates ($1 \times 10^5$ cells/well) and cultivated for 12 h. Cells were then infected with the viruses including rAAV-DCUg-NT, rAAV-DCUg-FL, rAAV-DM-NT, and rAAV-DM-FL at the dose of $1 \times 10^5$ vg per cell. The infected cells were cultured for 24 h and then incubated with or without 50 μg/mL FeNP. Cells were cultured for another 72 h, stained with AO&EB, and imaged by optical microscope (Olympus), and cell viability was evaluated using a CCK-8 assay (BS350B, Biosharp).

**Animal treatments.** Four-week-old BALB/c and 10-week-old C57BL/6J female mice were purchased from the Changzhou Cavens Laboratory Animal Co. Ltd (China). All animal experiments in this study followed the guidelines and ethics of the Animal Care and Use Committee of Southeast University (Nanjing, China). Tumor growth was monitored by volume measurement with calipers. Tumor volumes were calculated using formula $V = (ab^2)/2$, where a is the longest diameter and b is the shortest diameter. The mice were euthanized when the tumor size reached 2000 mm³ and various tissues (including heart, liver, spleen, lung, kidney, and tumor tissues) were collected for further analysis. Three animal models were performed.

Five batches of animal experiments were performed in the WEHI-3 xenografted model on BALB/c mice. WEHI-3 xenografts were generated by subcutaneously transplantation with $1 \times 10^7$ WEHI-3 cells into inner thighs. Mice were bred for 7 days for tumor formation. In the first batch of animal experiment, the tumor-bearing mice of WEHI-3 cell were randomly divided into six treatment groups (PBS, $n = 6$; FeNP, $n = 6$; rAAV-DCUg-NT, $n = 6$; rAAV-DCUg-NT + FeNP, $n = 6$; rAAV-DCUg-FL, $n = 6$; rAAV-DCUg-FL + FeNP, $n = 7$). The mice in the PBS group were then intravenously injected once with PBS. The mice in the rAAV-DCUg-NT and rAAV-DCUg-NT + FeNP groups were intravenously injected once with rAAV-DCUg-NT. The mice in the rAAV-DCUg-FL and rAAV-DCUg-FL + FeNP groups were intravenously injected once with rAAV-DCUg-FL. On the next day, the mice in the FeNP, rAAV-DCUg-NT + FeNP, and rAAV-DCUg-FL + FeNP groups were intravenously injected once with FeNP. The dosage of viruses and FeNP were $1 \times 10^{10}$ vg/mouse and 3 mg/kg body weight, respectively. Mice were euthanized and photographed on the seventh day post FeNP injection.

In the second batch of animal experiment, the tumor-bearing mice of WEHI-3 cell were randomly divided into five treatment groups (FeNP, $n = 6$; rAAV-DM-NT, $n = 6$; rAAV-DM-FL, $n = 7$; rAAV-DM-NT + FeNP, $n = 7$; rAAV-DM-FL + FeNP, $n = 6$). The mice were intravenously injected once with FeNP, rAAV-DM-NT, rAAV-DM-FL, rAAV-DM-NT + FeNP, and rAAV-DM-FL + FeNP, respectively. The dosage of viruses and FeNP were $1 \times 10^{10}$ vg/mouse and 3 mg/kg body weight, respectively. To simplify the drug administration, rAAV and FeNP were mixed together and intravenously injected to mice one time in this batch of animal experiment. Mice were euthanized and photographed on the seventh day post FeNP injection.

In the third batch of animal experiment, the tumor-bearing mice of WEHI-3 cell were randomly divided into three treatment groups (PBS, $n = 6$; rAAV-DM-NT + FeNP, $n = 6$; rAAV-DM-FL + FeNP, $n = 6$). The mice were intravenously injected three times every other day with PBS, rAAV-DM-NT + FeNP, and rAAV-DM-FL + FeNP, respectively. The dosage of viruses and FeNP were $1 \times 10^{10}$ vg/mouse and 3 mg/kg body weight, respectively. Virus and FeNP were injected as a mixture. Tumor size was measured every day. The Kaplan–Meier method was used to analyze the mice survival over time.

In the fourth batch of animal experiment, the tumor-bearing mice of WEHI-3 cell were randomly divided into three groups (rAAV-DM-NT + FeNP, rAAV-DM-FL + FeNP + lipro1 and rAAV-DM-FL + FeNP; $n = 6$). The mice were intravenously injected three times every other day with rAAV-DM-NT + FeNP (for the group of rAAV-DM-NT + FeNP) and rAAV-DM-FL + FeNP (for the groups of rAAV-DM-FL + FeNP + lipro1 and rAAV-DM-FL + FeNP), respectively. The dosage of viruses and FeNP were $2 \times 10^{10}$ vg/mouse and 3.6 mg/kg body weight, respectively. Virus and FeNP were injected as a mixture. Mice in the group of rAAV-DM-FL + FeNP + lipro1 were administered with liproxstatin-1 (10 mg/kg) once daily by intraperitoneal (i.p.) injection for 7 days. The body weight of the mice and tumor size were monitored daily. The mice were euthanized and photographed on the seventh day post first injection.

In the fifth batch of animal experiment, the tumor-bearing mice of WEHI-3 cell were randomly divided into eight groups (PBS-1, PBS-2, rAAV-DM-NT + FeNP-1, rAAV-DM-NT + FeNP-2, rAAV-DM-FL + FeNP + NAC-1, rAAV-DM-FL + FeNP + NAC-2, rAAV-DM-FL + FeNP-1, and rAAV-DM-FL + FeNP-2). The mice were intravenously injected three times every other day with PBS (PBS-1 and PBS-2), rAAV-DM-NT + FeNP (rAAV-DM-NT + FeNP-1 and rAAV-DM-NT + FeNP-2), and rAAV-DM-FL + FeNP (rAAV-DM-FL + FeNP + NAC-1, rAAV-DM-FL + FeNP + NAC-2, rAAV-DM-FL + FeNP-1, and rAAV-DM-FL + FeNP-2), respectively. The dosage of viruses and FeNP were $2 \times 10^{10}$ vg/mouse and 3.6 mg/kg body weight, respectively. Virus and FeNP were injected as mixture. Mice in the groups of rAAV-DM-FL + FeNP + NAC-1 and rAAV-DM-FL + FeNP + NAC-2 were administered NAC in their drinking water at 1 g/L. The

body weight of the mice and tumor size were monitored daily. The mice in the groups of PBS-1, rAAV-DM-NT + FeNP-1, rAAV-DM-FL + FeNP + NAC-1, and rAAV-DM-FL + FeNP-1 ($n = 9$) were euthanized and photographed on the seventh day post first injection. Blood and serum samples from each group were collected for routine blood test and serum biochemical parameter detection. Various tissues including heart, liver, spleen, lung, kidney, and tumor were harvested for H&E analysis and virus DNA and gene expression detections. The mice in the other four groups (PBS-2, rAAV-DM-NT + FeNP-2, rAAV-DM-FL + FeNP + NAC-2, and rAAV-DM-FL + FeNP-2; $n = 10$) were used for survival study.

CT-26 xenografts were generated on BALB/c mice by subcutaneously transplantation with $1 \times 10^6$ CT-26 cells into inner thighs. The mice were bred for 8 days for tumor formation. The tumor-bearing mice were randomly divided into four treatment groups (rAAV-DM-NT + FeNP-1, rAAV-DM-NT + FeNP-2, rAAV-DM-FL + FeNP-1, and rAAV-DM-FL + FeNP-2). Mice were intravenously administered three times every other day with rAAV-DM-NT + FeNP (rAAV-DM-NT + FeNP-1 and rAAV-DM-NT + FeNP-2) and rAAV-DM-FL + FeNP (rAAV-DM-FL + FeNP-1 and rAAV-DM-FL + FeNP-2), respectively. The dosage of viruses and FeNP were $2 \times 10^{10}$ vg/mouse and 3.6 mg/kg body weight, respectively. Virus and FeNP were injected as mixture. The body weight of the mice and tumor size were monitored daily. Mice in groups (rAAV-DM-NT + FeNP-1, rAAV-DM-FL + FeNP-1; $n = 10$) were euthanized and photographed on the tenth day post first injection. Blood and serum samples from each group were collected for routine blood test and serum biochemical parameter detection. Various tissues including heart, liver, spleen, lung, kidney, and tumor were harvested for H&E analysis and virus DNA and gene expression detection. The mice in the other two groups (rAAV-DM-NT + FeNP-2 and rAAV-DM-FL + FeNP-2; $n = 10$) were used for survival study.

Pulmonary metastatic melanoma model was established on C57BL/6J female mice by intravenously injection $2 \times 10^5$ B16F10 cells. The mice were bred for 10 days for tumor formation. The tumor-bearing mice were randomly divided into four treatment groups (rAAV-DM-NT + FeNP-1, rAAV-DM-NT + FeNP-2, rAAV-DM-FL + FeNP-1, and rAAV-DM-FL + FeNP-2). Mice were intravenously administered three times every other day with rAAV-DM-NT + FeNP (rAAV-DM-NT + FeNP-1 and rAAV-DM-NT + FeNP-2) and rAAV-DM-FL + FeNP (rAAV-DM-FL + FeNP-1 and rAAV-DM-FL + FeNP-2), respectively. The dosage of viruses and FeNP were $2 \times 10^{10}$ vg/mouse and 3.6 mg/kg body weight, respectively. Virus and FeNP were injected as mixture. The body weight of the mice was monitored daily. Mice in groups (rAAV-DM-NT + FeNP-1 and rAAV-DM-FL + FeNP-1; $n = 9$) were euthanized and photographed on the eleventh day post first injection. Blood and serum samples from each group were collected for routine blood test and serum biochemical parameter detection. Various tissues including heart, liver, spleen, lung, and kidney were harvested for H&E analysis and virus DNA and gene expression detection. The mice in the other two groups (rAAV-DM-NT + FeNP-2 and rAAV-DM-FL + FeNP-2; $n = 10$) were used for survival study. Mice were euthanized when the body weight loss was greater than 20% of the predosing weight.

For the safety assessment, ten BALB/c female mice were randomly divided into two treatment groups (PBS and rAAV-DM-FL + FeNP; $n = 5$). The mice were intravenously administered three times every other day with PBS and rAAV-DM-FL + FeNP, respectively. The dosage of virus and FeNP were $2 \times 10^{10}$ vg/mouse and 3.6 mg/kg body weight, respectively. Virus and FeNP were injected as mixture. The body weight of the mice was monitored daily. Mice were euthanized on the seventh day post first injection. Blood and serum samples from each group were collected for routine blood test and serum biochemical parameter detection. Various tissues including heart, liver, spleen, lung, and kidney were harvested for H&E analysis. The spleens were weighed and photographed.

**FeNC-based GIFT**. To investigate whether a PEI-modified Fe$_3$O$_4$ NP can be used as Fe nano-carriers (FeNC) for gene interference vectors, two FeNC-based GIFT experiments were performed, in which two batches of FeNC, named as FeNC-1 and FeNC-2, were used.

In the first FeNC-based GIFT experiment, various plasmids (including pDCUg-NT, pDCUg-FL, pDM-NT, and pDM-FL) were mixed with FeNC-1 (1 μg DNA/μg FeNC-1) to prepare plasmid DNA-loaded FeNC (FeNC@DNA), including FeNC-1@pDCUg-NT, FeNC-1@pDCUg-FL, FeNC-1@pDM-NT and FeNC-1@pDM-FL. Cells ($1 \times 10^5$) were seeded into 24-well plates and cultured overnight. Cells were then treated with FeNC, FeNC@DNA (totally 0.5 μg plasmid DNA), or plasmid DNA alone for 24 h. Cells were further cultured with a medium with or without 50 μg/mL FeNP for 72 h. Cells were stained with AO&EB at different time points (24 h, 48 h, and 72 h) and imaged.

In the second FeNC-based GIFT experiment, two plasmids (pDM-NT and pDM-FL) were mixed with FeNC-1 and FeNC-2 (1 μg DNA/μg FeNC) according to the manufacturer's instructions to prepare FeNC@DNA, including FeNC-1@pDM-FL and FeNC-2@pDM-FL. The prepared FeNC@DNA was used immediately to treat cells or left for 24 h before treating cells. Cells ($1 \times 10^5$) were seeded into 24-well plates and cultured overnight. Cells were then cultured with a fresh medium contained 50 μg/mL FeNC or FeNC@DNA for 72 h. Cells were stained with AO&EB at different time points (24 h, 48 h, and 72 h) and imaged.

To investigate the in vivo antitumor effect of FeNC-based GIFT, two batches of animal experiments were performed. In the first batch of animal experiment, the tumor-bearing mice of WEHI-3 cell were randomly divided into six treatment groups (PBS, $n = 6$; FeNC, $n = 6$; pAAV-DM-NT + FeNC, $n = 6$; pAAV-DM-FL + FeNC, $n = 7$; pAAV-DCUg-NT + FeNC, $n = 6$; and pAAV-DCUg-FL + FeNC, $n = 7$). Each group of mice was then intravenously injected with PBS, FeNC, pAAV-DM-NT + FeNC, pAAV-DM-FL + FeNC, pAAV-DCUg-NT + MT-FeNC, and pAAV-DCUg-FL + FeNC, respectively. Various plasmids and FeNC were mixed according to the manufacturer's instruction before injection, the dosage of different plasmids and FeNC were 2 mg/kg and 3 mg/kg body weight, respectively. Mice were euthanized and photographed on the seventh day post FeNP injection. In the second batch of animal experiments, the tumor-bearing mice of WEHI-3 cell were randomly divided into two treatment groups (pAAV-DM-NT + FeNC, $n = 5$; pAAV-DM-FL + FeNC; $n = 6$). The mice were intravenously administered two times every other day with pAAV-DM-NT + FeNC and pAAV-DM-FL + FeNC, respectively. The dosage of plasmid and FeNC per injection was same as above. The mice were euthanized on the seventh day post treatment. Tumor size was measured every day. Tumors were isolated and tumor weight were measured.

**H&E staining**. Tissues including heart, liver, spleen, lung, kidney and tumor were dissected, embedded in paraffin, sectioned, and stained with H&E using routine methods. Briefly, tissues were resected and fixed overnight in 4% paraformaldehyde solution (Sangon Biotech, China) at room temperature overnight. Subsequently, fixed specimens were embedded in paraffin, divided into 5-μm-thick sections, and then stained with hematoxylin staining solution (C0107, Beyotime) and eosin staining solution (C0109, Beyotime). The prepared slides were photographed by a microscope (IX51, Olympus).

**Quantitative PCR**. Total RNA was isolated from cell lines at 48 h post incubation with FeNP or mouse tissues using TRIzol™ (Invitrogen) according to the manufacturer's protocol. The complementary DNA (cDNA) was generated using the FastKing RT kit (TIANGEN) according to the manufacturer's instruction. The genomic DNA (gDNA) was extracted from various tissues of mice using the TIANamp Genomic DNA Kit (TIANGEN). Amplification for the genes of interest from cDNA and gDNA was performed by qPCR using the Hieff qPCR SYBR Green Master Mix (Yeasen). The primers used for qPCR are shown in the Supplementary Table 4. Triplicate samples per treatment were evaluated on an ABI Step One Plus (Applied Biosystems) and StepOne software. Relative mRNA transcript levels were compared to the *GADPH* internal reference and calculated as relative quantity (RQ) according to the following equation: RQ = $2^{-\Delta\Delta Ct}$. Virus DNA abundance were normalized to the *GADPH* internal reference and calculated according to the following equation: RQ = $2^{-\Delta Ct}$. Cas13a mRNA expression levels were showed as Ct values. All experiments were performed in triplicates and repeated a minimum of three times.

NF-κB RelA/p65 expression in cells were detected by quantitative PCR (qPCR) using the primers Human/Murine RELA-F/R and Human/Murine GAPDH-F/R. The results are normalized to *GAPDH* and analyzed by $2^{-\Delta Ct}$ method. All the qPCR primers were verified as being specific based on melting curve analysis and were listed in the Supplementary Table 4.

**Pharmacokinetics and biodistribution**. For pharmacokinetics study, four-week-old healthy BALB/c female mice ($n = 3$ for each group) were intravenously injected with rAAV-DM-FL + FeNP and pAAV-DM-FL + FeNC, respectively. The dosage of viruses and FeNP were $2 \times 10^{10}$ vg/mouse and 3.6 mg/kg body weight, respectively. The dosage of plasmids and FeNC were 2 mg/kg and 3 mg/kg body weight, respectively. The blood was collected at various times (1 min, 5 min, 10 min, 0.5 h, 1 h, 2 h, 4 h, 8 h, 12 h, and 24 h) after injection. For biodistribution study, the WEHI-3-tumor-bearing mice were randomly divided into three groups (PBS, rAAV-DM-FL + FeNP, pAAV-DM-FL + FeNC; $n = 9$). The mice were intravenously administered with corresponding reagents, respectively. The dosage of reagents was same as above. Mice were sacrificed, and the tumors and major organs were harvested at the predesignated time points (12 h, 24 h, and 48 h). The content of iron and rAAV/pAAV in tissues (blood, tumors and organs) were measured by ICP-MS and qPCR, respectively. The iron content in tissues of PBS group was used as the background value.

**Statistical analysis**. All data are presented as means values ± standard deviation (S.D.), and statistical analysis and graphs were performed with GraphPad Prism 8.0 software. Statistical differences between two groups were determined using two-tailed Student's t-test. Comparisons of three or more groups were determined by one-way or two-way analysis of variance (ANOVA) with Tukey's correction for multiple comparisons. The Kaplan–Meier method was used to analyze the differences in animal survival and the $P$ value was calculated by the log-rank test. Differences at $P < 0.05$ were considered statistically significant.

**Reporting summary**. Further information on research design is available in the Nature Research Reporting Summary linked to this article.

## Data availability

All data supporting the findings of this study are available within the article and its supplementary information files. Source data are provided with this paper.

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

## Acknowledgements

This work was supported by the National Key Research and Development Program of China (2017YFA0205502) and the National Natural Science Foundation of China (61971122).

## Author contributions

J.W. conceived and supervised the project. J.W. and J.G. designed the experiments. J.G. and T.L. performed main experiments and analyzed the data. J.W. and J.G. wrote the manuscript. All the authors discussed the results and commented on the manuscript.

## Competing interests

The authors declare no competing interests.
