## [Peer Review File · Nature Communications]

REVIEWER COMMENTS

Reviewer #1 (Remarks to the Author):

The authors have addressed most of my comments in an adequate way; however, one issue remains and requires additional experimentation. The authors used NAC in order to rescue some of the in vivo effects. Other than stated by the authors, NAC is not a bona fide ferroptosis inhibitor because it is just a small molecule compound to correct for intracellular cysteine levels. Increased level of cysteine can be used for GSH biosynthesis or can even be a substrate of GPX4 in the absence of GSH. In case of GPX4 loss or inhibited GPX4, NAC fails to protect against lipid peroxidation. Therefore, I strongly recommend to use an in vivo efficacious ferroptosis inhibitor (like Lip-1) to check whether it blunts some of the in vivo effects of the herein proposed “gene interfered-ferroptosis therapy”. This is even more important as some of the rescuing effects in cell culture are not particularly strong (see Fig. 4). Beyond this, I have no further objections.

Reviewer #4 (Remarks to the Author):

The authors reported on the combination therapy of the iron nanoparticle and knockdown of the genes that metabolize irons. For the gene knockdown, the authors used a AAV and PEI-coated iron particle that encapsulate pDNA.

From the point of view of the drug delivery system, the proof of concept is poor.

1. The pharmacokinetics and the tissue distribution of the particle and AAV was not clarified. It is plausible that simple PEI-coated particle (positively charged particle) rapidly eliminate form the blood, and accumulate to the liver, spleen and lung.

Especially, how much the particles accumulate to the tumor?

2. The in vivo gene knockdown effect was evaluated.

3. In relation to the comment 2, the gene knockdown effect in other tissues should be clarified.

4. It is not clear how the nanoparticle accumulates to the tumor. The particles accumulate to the tumor via EPR effect?

5. Current study indicated that the tumor accumulation of the nanoparticle via EPR effect generally overestimate the clinical situation: the accumulation of the particle to the tumor in patients are poor, while the accumulation was clearly observed in the mouse tumor-bearing model. As I comment in #4, the particle does not mount the targeting strategy for the tumor accumulation. I this strategy is not suit to be the translational study.

Others:

6. The authors used two-tailed Student's t-test for the statistical analysis as far as I read the material and methods. However, two-tailed Student's t-test is not allowed for the multiple comparison.

In summary, I think the manuscript does not suit to be accepted to the Nature communication.

REVIEWER COMMENTS

Reviewer #1 (Remarks to the Author):

The authors have addressed most of my comments in an adequate way; however, one issue remains and requires additional experimentation. The authors used NAC in order to rescue some of the in vivo effects. Other than stated by the authors, NAC is not a bona fide ferroptosis inhibitor because it is just a small molecule compound to correct for intracellular cysteine levels. Increased level of cysteine can be used for GSH biosynthesis or can even be a substrate of GPX4 in the absence of GSH. In case of GPX4 loss or inhibited GPX4, NAC fails to protect against lipid peroxidation. Therefore, I strongly recommend to use an in vivo efficacious ferroptosis inhibitor (like Lip-1) to check whether it blunts some of the in vivo effects of the herein proposed “gene interfered-ferroptosis therapy”. This is even more important as some of the rescuing effects in cell culture are not particularly strong (see Fig. 4). Beyond this, I have no further objections.

Response: Yes. Thank you for this precious comment and suggestion. As you suggested, we have performed an additional animal experiment to evaluate whether liproxstatin-1 can blunt the in vivo effects of GIFT (gene interfered-ferroptosis therapy). The results indicate that intraperitoneal injection of liproxstatin-1 (10 mg/kg) once daily significantly relieved tumor suppression made by rAAV-DM-FL+FeNP in mice (Supplementary Fig. 50). The related methods and results were added to the revised manuscript.

Reviewer #4 (Remarks to the Author):

The authors reported on the combination therapy of the iron nanoparticle and knockdown of the genes that metabolize irons. For the gene knockdown, the authors used a AAV and PEI-coated iron particle that encapsulate pDNA.

From the point of view of the drug delivery system, the proof of concept is poor.

1. The pharmacokinetics and the tissue distribution of the particle and AAV was not clarified. It is plausible that simple PEI-coated particle (positively charged particle) rapidly eliminate form the blood, and accumulate to the liver, spleen and lung. Especially, how much the particles accumulate to the tumor?

Response: Thank you for this precious comment. We performed additional animal experiments to evaluate the pharmacokinetics and biodistribution of rAAV, FeNP (DMSA@Fe₃O₄), pAAV and FeNC (PEI@Fe₃O₄). The results were shown as the Supplementary Fig. 63. The resultd showed that the half-life time of rAAV, FeNP, pAAV and FeNC was 11.1 h (Supplementary Fig. 47a), 5.0 h (Supplementary Fig. 47b), 5.2 h (Supplementary Fig. 47c), and 4.9 h (Supplementary Fig. 47d), respectively. The results also showed that all the reagents could be highly accumulated to the liver and spleen due to EPR effect; however, except the two organs of reticuloendothelial system, these materials were clearly accumulated in tumor. The results indicated that 29.5% injected dose per gram of tissue (ID g⁻¹) of rAAV, 24.3% (ID g⁻¹) of FeNP, 20.4% (ID g⁻¹) of pAAV, and 23.4% (ID g⁻¹) of FeNC were accumulated in the tumors at 24 h post injection (Supplementary Fig. 63). Please see the section “Pharmacokinetics and biodistribution” of the revised manuscript.

We also detected the iron content in tumors and major organs in the WEHI-3 xenografted mice. The results indicated that the rAAV-DM-FL+FeNP treatment only significantly increased the iron content of tumors (Supplementary Fig. 51a), but not other tissues (Supplementary Fig. 51b). Please note that the Supplementary Fig. 51b was newly added result.

The similar result was also obtained in the WEHI-3 xenografted mice treated with pAAV-DM-FL+FeNC (Supplementary Fig. 62k). Please note that the Supplementary Fig. 62k was newly added result.

The results of Supplementary Fig. 51a and b and Supplementary Fig. 62k indicated that the GIFT treatment could specifically and significantly increase the iron content of tumor.

2. The in vivo gene knockdown effect was evaluated.

3. In relation to the comment 2, the gene knockdown effect in other tissues should be clarified.

Response: Thank you for pointing this out. The responses to question 2 and 3 are as follows.

In the old manuscript, we in fact detected the in vivo gene knockdown effect in tumors and major organs in the animal experiments. Reviewer may not notice the related content.

We detected the in vivo gene knockdown effect in tumors and major organs in the WEHI-3 xenografted mice. It was shown in the second paragraph of section “Virus-based GIFT antitumor in vivo” of the manuscript as follows:

To further characterize the in vivo antitumor effect of GIFT, we detected the abundance of GIV DNA, and mRNA of effector gene Cas13a and two target genes FPN and LCN2 in various tissues by qPCR. The results showed that the GIV DNA distributed in all detected tissues, especially in tumor (Supplementary Fig. 48e), and Cas13a was only expressed in tumors (Supplementary Fig. 48f). The expression of two target genes (FPN and LCN2) was only significantly knocked down in tumors (Supplementary Fig. 48g,h).

It was also explained in the third paragraph of section “Virus-based GIFT antitumor in vivo” of the manuscript as follows:

To further characterize the treatment, we also detected the iron content and the abundance of DNA of GIV and mRNA of RELA, FPN and LCN2 in various tissues. The results indicated that the rAAV-DM-FL+FeNP treatment significantly increased the iron content of tumors (Supplementary Fig. 51a), but not other tissues (Supplementary Fig. 51b). The GIV DNA distributed in all detected tissues, especially in tumor (Supplementary Fig. 51c). RelA was only highly expressed in tumors (Supplementary Fig. 51d) and two target genes FPN and LCN2 were only significantly knocked down in tumor (Supplementary Fig. 51e,f).

We also detected the in vivo gene knockdown effect in tumors and major organs in the CT-26 xenografted mice. It was explained in the third paragraph of section “Virus-based GIFT antitumor in vivo” of the manuscript as follows:

Detections of iron content of tumors (Supplementary Fig. 53a), abundance of rAAV DNA (Supplementary Fig. 53b), and mRNA of RELA (Supplementary Fig. 53c), FPN (Supplementary Fig. 53d), and LCN2 (Supplementary Fig. 53e) obtained the similar results that were observed in the WEHI-3 model.

It was also explained in the second paragraph of section “FeNC-based GIFT antitumor in vitro and in vivo” of the manuscript as follows:

Moreover, the transcription of two target genes FPN and LCN2 were only significantly knocked down in tumors (Supplementary Fig. 62e,f).

Please see the related figures and the revised manuscript.

4. It is not clear how the nanoparticle accumulates to the tumor. The particles accumulate to the tumor via EPR effect?

5. Current study indicated that the tumor accumulation of the nanoparticle via EPR effect generally overestimate the clinical situation: the accumulation of the particle to the tumor in patients are poor, while the accumulation was clearly observed in the mouse tumor-bearing model. As I comment in #4, the particle does not mount the targeting strategy for the tumor accumulation. I this strategy is not suit to be the translational study.

Response: Thank you for this precious comment and suggestion. The reponses to question 4 and 5 are as follows.

In this study, we used two kinds of iron nanoparticles, DMSA-coated Fe₃O₄ nanoparticles (FeNP) and PEI-coated Fe₃O₄ nanoparticles (FeNC). We detected the in vivo iron contents both in tumors and major organs in two tumor models (WEHI-3 and CT-26). The results indicated that the treatments of both nanoparticles resulted in significantly increase of iron contents in tumor, indicating that nanoparticles were accumulated in tumors. Because these nanoparticles were not modified with any tumor-specific ligands, we thought that they were accumulated in tumors by passive targeting. This was supported by a latest research, which revealed that nanoparticle can accumulate to the tumor via passive EPR effect and active transport through trans-endothelial pathways, and the latter accounts for the majority of nanoparticle accumulation in tumors (Sindhwani, S. et al. The entry of nanoparticles into solid tumours. *Nature Materials* 2020, 19:566-575).

In this study, we used two nanoparticles that were not modified with any tumor-specific ligands. Therefore, we can treated three different kinds of tumors with FeNP. At present, little highly specific tumor biomarkers on cell surface were identified, especially solid tumors. Therefore, preparing nanoparticles with highly tumor specificity is still difficult. On the other hand, if a targeting ligand specific to a kind of tumor were modified on nanoparticles, this kind of nanoparticles may be unsuitable for treating other tumors because different tumors having different target (Peng XH, et al. Targeted magnetic iron oxide nanoparticles for tumor imaging and therapy. *Int J Nanomedicine*. 2008;3(3):311-21), which will greatly complicate the preparation of nanoparticles for traeting different tumors. Therefore, the nanoparticles used in this study could be universally used to treat various tumors as shown by this study.

At present, we can only investigated the GIFT with mice bearing different kinds of tumors. We are sorry not to be able to treat tumors on human with GIFT now. The exact antitumor effect of GIFT on human has to be investigated in future if this treatment can be permitted to clinical trial. However, iron oxide nanoparticles are the only approved metallic nanoparticles for clinical use and clinical applications are likely to be limited to only iron oxide nanoparticles because of their demonstrated safety. Iron oxide nanoparticles have enjoyed clinical use for about nine decades demonstrating safety, and considerable clinical utility and versatility. FDA-approved applications of iron oxide nanoparticles include cancer diagnosis, cancer hyperthermia therapy, and iron deficiency anemia (Soetaert F, at el. Cancer therapy with iron oxide nanoparticles: Agents of thermal and immune therapies. *Adv Drug Deliv Rev*. 2020;163-164:65-83). Of cause, as reviwer pointed out the most significant challenge with systemic delivery of magnetic iron oxide nanoparticles perhaps is achieving sufficient concentration in tumors. This

key issue has to be investigated in the future with the struggle of the whole scientific field of nanomaterials. By detecting iron contents of tumor and various organs, this study revealed that that iron nanoparticles can be accumulated in tumor and the GIFT treatment significantly increased the iron contents of tumors but not other tissues. Thank you.

Others:

6. The authors used two-tailed Student's t-test for the statistical analysis as far as I read the material and methods. However, two-tailed Student's t-test is not allowed for the multiple comparison.

Response: Yes. Thank you for pointing this out. We re-performed the statistical analysis. Statistical differences between two groups were determined using two-tailed Student's t-test. The differences between multiple groups were analyzed using one-way or two-way analysis of variance (ANOVA) with Tukey's correction for multiple comparisons. Please see the revised manuscript.

REVIEWERS' COMMENTS

Reviewer #1 (Remarks to the Author):

The authors have performed the requested experiment and as such I have no further comments.

Reviewer #4 (Remarks to the Author):

The authors corrected the manuscript considering the major part of my comment. However, I still think that the authors should add some discussion on the EPR effect. The authors should add some comment in the main text since these are quite important aspect.

Reviewer #1 (Remarks to the Author):

The authors have performed the requested experiment and as such I have no further comments.

Reviewer #4 (Remarks to the Author):

The authors corrected the manuscript considering the major part of my comment. However, I still think that the authors should add some discussion on the EPR effect. The authors should add some comment in the main text since these are quite important aspect.

Response: Yes, as you pointed out the EPR effect is important to the therapy method we developed in this study, especially to iron nanoparticles. We therefore added some discussions in main text about EPR. Please see the final paragraph of the Results and discussion section. Thank you.